# Winner-Take-All Column Row Sampling for Memory Efficient Adaptation of Language Model

**Zirui Liu**[1*], **Guanchu Wang**[1*], **Shaochen Zhong**[1], **Zhaozhuo Xu**[2], **Daochen Zha**[1],
**Ruixiang Tang**[1], **Zhimeng Jiang**[3], **Kaixiong Zhou**[1], **Vipin Chaudhary**[4], **Shuai Xu**[4], **Xia Hu**[1]
[1]Rice University, [2]Stevens Institute of Technology, [3]Texas A&M University,
[4]Case Western Reserve University
{zl105,gw22,hz88,daochen.zha,ruixiang.tang,kaixiong.zhou,xia.hu}@rice.edu;
zxu79@stevens.edu; zhimengj@tamu.edu;
{vipin, sxx214}@case.edu

## Abstract

As the model size grows rapidly, fine-tuning the large pre-trained language model has become increasingly difficult due to its extensive memory usage. Previous works usually focus on reducing the number of trainable parameters in the network. While the model parameters do contribute to memory usage, the primary memory bottleneck during training arises from storing feature maps, also known as activations, as they are crucial for gradient calculation. Notably, machine learning models are typically trained using stochastic gradient descent. We argue that in stochastic optimization, models can handle noisy gradients as long as the gradient estimator is unbiased with reasonable variance. Following this motivation, we propose a new family of unbiased estimators called `WTA-CRS`, for matrix production with reduced variance, which only requires storing the sub-sampled activations for calculating the gradient. Our work provides both theoretical and experimental evidence that, in the context of tuning transformers, our proposed estimators exhibit lower variance compared to existing ones. By replacing the linear operation with our approximated one in transformers, we can achieve up to $2.7\times$ peak memory reduction with almost no accuracy drop and enables up to $6.4\times$ larger batch size. Under the same hardware, `WTA-CRS` enables better down-streaming task performance by applying larger models and/or faster training speed with larger batch sizes. The code is available at `https://github.com/zirui-ray-liu/WTACRS/`.

## 1 Introduction

Pre-trained language models (LMs) with transformer architecture have achieved remarkable success in numerous natural language processing (NLP) tasks [1, 2, 3, 4, 5]. Specifically, these models are trained on vast text corpora to acquire general-purpose representations, which are then adapted to a specific task by fine-tuning on task-specific data. In recent studies, it has been convincingly demonstrated that significantly increasing the number of parameters in pre-trained LMs leads to remarkable improvements in performance [6, 7]. As a result, there is now an urgent necessity to effectively adapt these models, equipped with billion-scale parameters, to a wide range of tasks.

However, a significant disparity exists between the memory requirements of pre-trained LMs and the capacity of current hardware, particularly GPUs. For example, even a GPU with 24GB memory cannot accommodate the fine-tuning process of the T5-3B model [3] with batch size one, which boasts three billion parameters. Without additional techniques, attempting to fine-tune billion-scale LMs on a single GPU is impossible. Although model-parallel fine-tuning is feasible, the majority of the time, we cannot bear the expense of acquiring multiple GPUs or the communication overhead

---

*Equal contribution. The order of authors is determined by flipping a coin.

involved. To ensure the smooth deployment of language models during the fine-tuning process, it is crucial to adapt them for operation on a single GPU.

To address this issue, several parameter-efficient tuning methods are proposed [8, 9, 10, 11, 12, 13, 14]. Specifically, adapters [14, 13] insert a small module into the transformer blocks and only update it while keeping other parameters fixed. Similarly, prompt tuning [8] introduces a small vector that is concatenated with the input embeddings and updated during the tuning process. LoRA [12] injects trainable rank decomposition matrices into the transformer block, updating them while freezing the others. Parameter-efficient tuning methods mainly reduce the memory taken by the optimizer states [15, 12]. Although the optimizer states contribute to the memory footprint, *storing activations (or feature maps) is the main memory bottleneck during training* (often $> 70\%$) [16, 17, 18, 19]. Thus, parameter-efficient methods often do not reduce memory usage by much [9, 19].

In parallel, we can reduce the main memory bottleneck by reducing the activation storage in fine-tuning. Since transformer-based models are mainly built based on the linear layer, a less-explored direction is to replace the expensive matrix multiplication operation with its memory-efficient estimations using column-row sampling (CRS) [21, 22]. The key idea of CRS is to sub-sample tensors onto low-dimensional spaces and perform the original operations here. Specifically, for the linear operation between two matrices $\mathbf{A} \in \mathbb{R}^{n \times m}$ and $\mathbf{B} \in \mathbb{R}^{m \times q}$ (in the context of machine learning, $\mathbf{A}$ is often activations), **we first sample $k$ ($k < m$) column-row pairs according to a pre-defined distribution.** Then we obtain $\mathbf{A}' \in \mathbb{R}^{n \times k}$ and $\mathbf{B}' \in \mathbb{R}^{k \times q}$ ($k < m$) by picking $k$ columns of $\mathbf{A}$ and the corresponding rows of $\mathbf{B}$ according to the sampled column-row pairs [22]. Finally, we estimate $\mathbf{AB} \approx \mathbf{A}'\mathbf{B}'$. In this way, we only need to store the sub-matrix $\mathbf{A}'$ and $\mathbf{B}'$ in GPU memory to perform the computation. Moreover, transformer-based models training/tuning are performed with the first-order stochastic optimizer, e.g., Adam [15]. In stochastic optimization, models can work with noisy gradients, *as long as the gradient estimator is unbiased and has a reasonable variance.* In view of such, we ask: **why spend resources on obtaining exact gradients when we are using stochastic optimization?** Motivated by this, we focus on obtaining unbiased gradients cheaply with approximated matrix multiplication.

The approximation method reduces the memory usage at the cost of giving outputs with variance. Thus there naturally exists an accuracy-memory trade-off. The main challenge is how to integrate the approximated matrix multiplication into transformer with minimal gradient variance. In this paper, we propose a new family of unbiased estimator for matrix multiplication with reduced variance, dubbed Winner-Take-All Column-Row Sampling (`WTA-CRS`). Compared to CRS, `WTA-CRS` reduces the variance of an estimator by focusing more on high-probability regions of the sampling distribution. Moreover, `WTA-CRS` can serve as a drop-in replacement for the linear operation in transformers, providing an unbiased weight gradient with reduced memory usage. As shown in Figure 1, our method achieves better accuracy-memory trade-off than state-of-the-art memory-efficient tuning methods, e.g., LST [9] and LoRA [12]. Moreover, since `WTA-CRS` executed at the operation level, it is orthogonal to most of the existing parameter-efficient tuning methods. Our contributions are highlighted as follows:

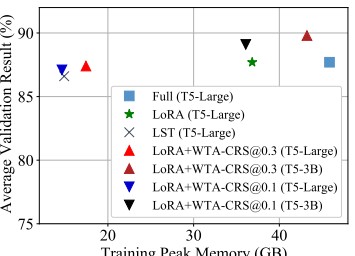

Fig. 1. Accuracy-memory trade-off of `WTA-CRS` and other memory-efficient tuning methods. Unless specially stated, we use the T5-Large in the figure.

- We design a new family of unbiased estimator for matrix multiplication with reduced variance. We theoretically and experimentally verify that it has smaller variance than the established one under the context of tuning transformer.
- By replacing the linear operation with `WTA-CRS` in transformers, we can achieve up to $2.7\times$ peak memory reduction with almost no accuracy drop, and enables up to $6.4\times$ larger batch size. For example, to fully tune T5-3B, it requires 66.4GB of memory for in a mini-batch size of 32, which relies on A100 GPUs with a capacity of 80GB. With `WTA-CRS`, it only requires 21.6GB of memory for fine-tuning, which can run on a GPU with 24GB memory, e.g. RTX3090Ti.
- We implement `WTA-CRS` as a ready-to-use extension for Pytorch with an easy-to-use API that can also be combined with other memory-saving techniques.

## 2 Background and Preliminary

In this section, we first analyze the memory usage of transformers. Then we introduce the background on the approximated matrix multiplication.

### 2.1 The Memory Usage of Transformers

In each training step of backpropagation, it has exactly two phases, i.e., one forward phase and one backward phase. Transformer-based models are mainly built based on the linear operation, which can be written as:

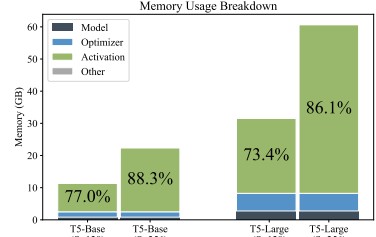

Fig. 2. The GPU memory usage breakdown for fine-tuning T5 [23], where the batch size $B$ is 64 and sequential length $S$ is 128 or 256.

$$\text{Forward Pass} \quad \mathbf{Z} = \texttt{GEMM}(\mathbf{H}, \mathbf{W}), \tag{1a}$$

$$\text{Backward Pass} \quad \nabla\mathbf{H} = \texttt{GEMM}(\nabla\mathbf{Z}, \mathbf{W}^\top), \tag{1b}$$

$$\nabla\mathbf{W} = \texttt{GEMM}(\mathbf{H}^\top, \nabla\mathbf{Z}), \tag{1c}$$

where $\texttt{GEMM}(\cdot, \cdot)$ is the General Matrix Multiplication operation, $\mathbf{H}$ and $\mathbf{Z}$ are the activation (or input feature maps) and output feature maps, respectively. $\mathbf{W}$ is the weight of the linear layer. $\nabla\mathbf{H}$, $\nabla\mathbf{W}$, and $\nabla\mathbf{Z}$ are the gradient of $\mathbf{H}$, $\mathbf{W}$, and $\mathbf{Z}$, respectively. From Equation (1c), activations $\mathbf{H}$ are used in the backward phase. In commonly used deep learning framework [24, 25], it requires storing $\mathbf{H}$ in GPU memory during the forward pass, for calculating the weight gradient $\nabla\mathbf{W}$ in the backward pass.

Previous works show that although the model parameters contribute to the memory footprint, activations (e.g., storing $\mathbf{H}$) are the main memory bottleneck during training [16, 17, 18, 19]. To get a sense of the scale, we show in Figure 2 that for popular transformer models like T5, activations may take roughly $73 \sim 88\%$ of the total memory, depending on the batch size $B$ and sequential length $S$.

### 2.2 Approximated GEMM With Sampling

Let $\mathbf{X} \in \mathbb{R}^{n \times m}$, $\mathbf{Y} \in \mathbb{R}^{m \times q}$ be two matrices. The goal is to efficiently estimate the matrix production $\mathbf{XY}$. Singular Value Decomposition (SVD) outputs provably optimal low-rank estimation of $\mathbf{XY}$ [21]. However, SVD is almost as expensive as matrix production itself. Instead, the sampling algorithm is proposed to approximate the matrix product $\mathbf{XY}$ by sampling $k$ columns of $\mathbf{X}$ and corresponding rows of $\mathbf{Y}$ to form smaller matrices, which are then multiplied as usual [22, 26]:

$$\texttt{GEMM}(\mathbf{X}, \mathbf{Y}) = \sum_{i=1}^{m} \mathbf{X}_{:,i} \mathbf{Y}_{i,:} \approx \sum_{t=1}^{k} \frac{1}{k p_{i_t}} \mathbf{X}_{:,i_t} \mathbf{Y}_{i_t,:} = \mathbf{X}'\mathbf{Y}', \tag{2}$$

where $\mathbf{X}_{:,i} \in \mathbb{R}^{n \times 1}$ and $\mathbf{Y}_{i,:} \in \mathbb{R}^{1 \times q}$ are the $i^{\text{th}}$ column and row of $\mathbf{X}$ and $\mathbf{Y}$, respectively. **In this paper, we call $(\mathbf{X}_{:,i}, \mathbf{Y}_{i,:})$ the $i^{\text{th}}$ column-row pair.** $k$ is the number of sampled pairs ($1 \leq k \leq m$). $\mathcal{P} = \{p_i\}_{i=1}^{m}$ is a probability distribution over the column-row pairs. $i_t \in \{1, \cdots m\}$ is the index of the sampled column-row pair at the $t^{\text{th}}$ trial. $s_t$ is the scale factor. $\mathbf{X}' \in \mathbb{R}^{n \times k}$ and $\mathbf{Y}' \in \mathbb{R}^{k \times q}$ are the normalized sub-matrices sliced according to the sampled column-row pairs.

Existing work [22] shows $\mathbf{X}'\mathbf{Y}'$ is an unbiased estimation of $\mathbf{XY}$, i.e., $\mathbb{E}[\mathbf{X}'\mathbf{Y}'] = \mathbf{XY}$. Furthermore, the approximation error $\mathbb{E}[||\mathbf{XY} - \mathbf{X}'\mathbf{Y}'||_F]$ is minimized when the probabilities $\{p_i\}_{i=1}^{m}$ are proportional to the product of the column-row Euclidean norms [22] (Proof in Appendix C):

$$p_i = \frac{||\mathbf{X}_{:,i}||_2 \, ||\mathbf{Y}_{i,:}||_2}{\sum_{j=1}^{m} ||\mathbf{X}_{:,j}||_2 \, ||\mathbf{Y}_{j,:}||_2}. \tag{3}$$

As we analyzed in Section 2.1, storing the activation $\mathbf{H}$ is the major memory bottleneck. **If we can replace $\texttt{GEMM}(\mathbf{H}^\top, \nabla\mathbf{Z})$ in Equation (1c) with $\mathbf{H}'^\top \nabla\mathbf{Z}'$ following the paradigm of Equation (2), then we only need $\mathbf{H}'$ instead of $\mathbf{H}$ in GPU memory to compute the gradient, which significantly decreases the memory usage of activations.** This estimation linearly reduces the memory complexity from $\mathcal{O}(nm)$ to $\mathcal{O}(nk)$. Also, the total number of floating point operations (FLOPs) is reduced as

well since the computation is executed on two smaller matrices. *For the ease of illustration, in this paper we call the distribution in Equation (3) the **column-row index distribution***. In the next section, we explore how to reduce memory usage via sampling-based matrix multiplication.

## 3 Methodology

In recent years, we have observed that deep neural network training can be performed almost entirely with first-order *stochastic optimization* [15]. Thus intuitively, *in stochastic optimization we can reduce the resources spent on obtaining gradients, as long as the estimated gradient is unbiased with reasonable variance* [27, 28, 29]. Following this motivation, we first design a new unbiased estimator for matrix multiplication with reduced variance compared to the one in Equation (2) (Section 3.1 ). Then we introduce how to replace the GEMM in Transformer with its approximated version to reduce the memory usage (Section 3.2).

### 3.1 Winner-Take-All Column-Row Sampling: A New Unbiased Estimator for GEMM

In this section, we mathematically design a new unbiased estimator for GEMM with reduced variance called WTA-CRS (Winner-Take-All Column-Row Sampling). Following the notation in Section 2.2, let $\mathbf{X} \in \mathbb{R}^{n \times m}$, $\mathbf{Y} \in \mathbb{R}^{m \times q}$ be two matrices. $\mathcal{P} = \{p_i\}_{i=1}^m$ is the column-row index distribution in Equation (3)[2]. We first define the variable $f(i)$ as $f(i) = \frac{\mathbf{X}_{:,i}\mathbf{Y}_{i:}}{p_i}$.

$f(i)$ is an unbiased estimation for the matrix production between $\mathbf{X}$ and $\mathbf{Y}$. To see this,

$$\mathbb{E}_{j \sim \mathcal{P}}[f(j)] = \sum_{i=1}^m p_i \frac{\mathbf{X}_{:,i}\mathbf{Y}_{i:}}{p_i} = \mathbf{X}\mathbf{Y}.$$

We note that the prior approximated matrix multiplication in Equation (2) is the direct extension of $f(i)$ by taking the average of $\{f(i_t)\}_{t=1}^k$ among $k$ independent random trials to reduce the variance. Here we explore an alternative approach to reduce the variance of $f(i)$ beyond simple averaging. Our core idea is to partition the column-row index distribution $\mathcal{P} = \{p_i\}_{i=1}^m$ into two complementary regions based on the probability mass: a high-probability region $\mathcal{P}^{\mathcal{C}}$ and a low-probability region $\mathcal{P}^{\mathcal{D} \setminus \mathcal{C}}$, where $\mathcal{D} = \{1, \cdots, m\}$ is the whole set and $\mathcal{C}$ is the set of the column-row index with the largest probability. **Let $\mathcal{C}$ be the set of column-row pair indices associated with $|\mathcal{C}|$ largest $p_i$.** We define WTA-CRS estimator for $\mathbf{X}\mathbf{Y}$ as follows:

$$\mathbb{E}_{j \sim \mathcal{P}^{\mathcal{D} \setminus \mathcal{C}}} \Big[ \sum_{c \in \mathcal{C}} f(c)p_c + (1 - \sum_{c \in \mathcal{C}} p_c)f(j) \Big]. \tag{4}$$

We note that **the random variable in Equation (4) is the column-row pair index $j$, and is only sampled from $\mathcal{D} \setminus \mathcal{C}$.** The estimator defined in Equation (4) contains two parts. The first part $\sum_{c \in \mathcal{C}} f(c)p_c$ has no relationship with the random variable $j$ and is summed deterministically. The second part $f(j)$ is sampled stocastically, but scaled by the factor $(1 - \sum_{c \in \mathcal{C}} p_c)$. When $\mathcal{P} = \{p_i\}_{i=1}^m$ is concentrated on a small number of atoms, the scaling factor $(1 - \sum_{c \in \mathcal{C}} p_c)$ for the stochastic term should be small. Therefore, we intuitively expect the estimator to have a small variance in this case due to a small scaling factor. In this way, we reduce the variance of an estimator by focusing more on high-probability regions of the distribution (winner-take-all). Below we formalize this intuition by showing the statistical property of our estimator regarding the bias and variance, respectively.

**Theorem 1** (Proof in Appendix C.2). *The estimator defined in Equation (4) is an unbiased estimator for matrix production $\mathbf{X}\mathbf{Y}$, i.e, $\mathbb{E}_{j \sim \mathcal{P}^{\mathcal{D} \setminus \mathcal{C}}}[\sum_{c \in \mathcal{C}} f(c)p_c + (1 - \sum_{c \in \mathcal{C}} p_c)f(j)] = \mathbf{X}\mathbf{Y}$.*

Theorem 1 states that our proposed estimator in Equation (4) is unbiased. Below we compare our proposed estimator to the CRS estimator in Equation (2) in terms of the variance. Suppose we have the budget of only utilizing $k$ column-row pairs for approximating the matrix production. From the implementation perspective, the estimator defined in Equation (2) estimates GEMM($\mathbf{X}, \mathbf{Y}$) as:

$$\text{(CRS)} \quad g(\mathbf{X}, \mathbf{Y}) = \frac{1}{k} \sum_{t=1}^k f(i_t), \quad i_1, \cdots, i_k \overset{\text{i.i.d}}{\sim} \mathcal{P}. \tag{5}$$

---

[2]Here we note that the theoretical analysis in this section can be applied to any probability distribution, not only limited to the one in Equation (3).

Our estimator defined in Equation (4) splits the budget $k$ into two parts. Namely, the first part explicitly sums the expectation terms for the largest probability group $\mathcal{C}$ ($|\mathcal{C}| < k$), while stochastically average $k - |\mathcal{C}|$ samples drawn from $\mathcal{D}\backslash\mathcal{C}$ to estimate the remaining terms, up to scale:

$$(\texttt{WTA-CRS}) \quad \hat{g}(\mathbf{X}, \mathbf{Y}) = \sum_{c \in \mathcal{C}} f(c)p(c) + \frac{1 - \sum_{c \in \mathcal{C}} p_c}{k - |\mathcal{C}|} \sum_{j=1}^{k-|\mathcal{C}|} f(j), \quad i_1, \cdots, i_{k-|\mathcal{C}|} \overset{\text{i.i.d}}{\sim} \mathcal{P}^{\mathcal{D}\backslash\mathcal{C}}. \quad (6)$$

**Theorem 2** (Proof in Appendix C.3). *Suppose the total budget of column-row pairs is $k$. If $\mathcal{C}$ satisfies*

$$\sum_{c \in \mathcal{C}} p_c > \frac{|\mathcal{C}|}{k}, \quad (7)$$

*then we have* $\mathrm{Var}[\hat{g}(\mathbf{X}, \mathbf{Y})] < \mathrm{Var}[g(\mathbf{X}, \mathbf{Y})]$. *Moreover,* $\mathrm{Var}[\hat{g}(\mathbf{X}, \mathbf{Y})]$ *is minimized when* $|\mathcal{C}| = \min_{|\mathcal{C}| \in \{0, \cdots, k\}} \frac{1 - \sum_{c \in \mathcal{C}} p_c}{k - |\mathcal{C}|}$.

Both the left- and right-hand sides of Equation (7) depend on the size of the highest probability group $|\mathcal{C}|$, which controls the number of high probability column-row pairs that are directly added without sampling. Below we experimentally investigate whether Equation (7) holds under the context of fine-tuning the transformer-based model with varying $|\mathcal{C}|$.

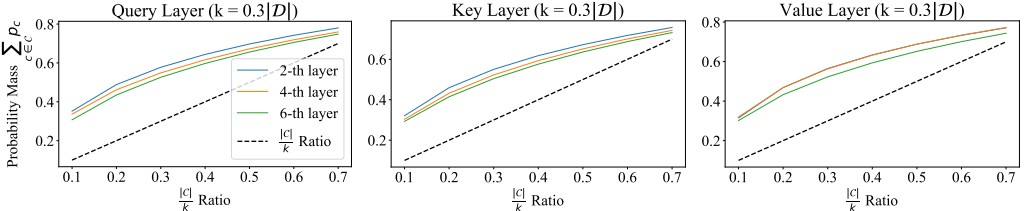

Fig. 3. The probability mass $\sum_{c \in \mathcal{C}} p_c$ versus $\frac{|\mathcal{C}|}{k}$ in Equation (7) at $k = 0.3|\mathcal{D}|$. Here we visualize the column-row index distribution of query/key/value projection layer in the T5-base model, which is fine-tuned on RTE dataset. More similar results can be found in Appendix E.1.

**Experimental analysis.** As shown in Figure 3, we visualize the two terms in Equation (3) for the column-row index distribution of query, key, and value projection in the self-attention module, respectively [1]. Specifically, we fix the total column-row pair budget $k = 0.3|\mathcal{D}|$ and change the size of the highest probability group $|\mathcal{C}|$ from 0 to $k$. We conclude that Equation (7) holds for most of the layers when fine-tuning transformers. Thus, we expect our `WTA-CRS` has better performance than CRS for adapting transformer-based models, which is later experimentally verified in Section 5.

### 3.2 Compress `GEMM` in Transformers with `WTA-CRS`

Previous work has shown that unbiasedness of the estimated gradient is crucial for the proper convergence of stochastic gradient descent [27, 28, 16, 30]. As shown in Section 2.1, we have three `GEMM` in the linear layer. Below we investigate how to replace `GEMM` with its approximated version in a way that the estimated gradient is unbiased.

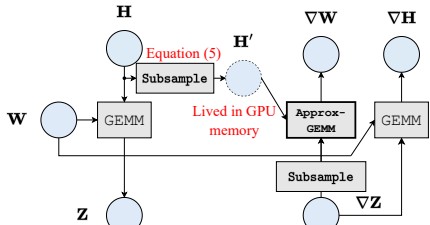

Fig. 4. The illustration of how to deploy `WTA-CRS` to linear layers. We only replace `GEMM` in Equation (1c) with its approximated version using `WTA-CRS`. The pseudocode is given in Appendix D Algorithm 1.

**Unbiasedness.** Previous work has shown that to ensure the unbiasedness of the gradient, the approximation can only be applied during the backward pass [16, 31, 21]. The rationale behind this conclusion is that we have $\mathbb{E}[f(x)] \neq f(\mathbb{E}[x])$ for any non-linear function $f(\cdot)$, e.g., $\mathbb{E}[x^2] \neq \mathbb{E}^2[x]$. Thus if we replace the forward `GEMM` in Equation (1a), even when the approximation method gives an unbiased estimation, i.e., $\mathbb{E}[\hat{g}(\mathbf{H}, \mathbf{W})] = \mathbf{H}\mathbf{W} = \mathbf{Z}$, the output activations (e.g., GeLU($\mathbf{Z}$)) are still biased since the activation function is non-linear, namely,

$$\text{GeLU}(\hat{g}(\mathbf{H}, \mathbf{W})) = \text{GeLU}(\mathbb{E}[\mathbf{Z}]) \neq \mathbb{E}[\text{GeLU}(\mathbf{Z})].$$

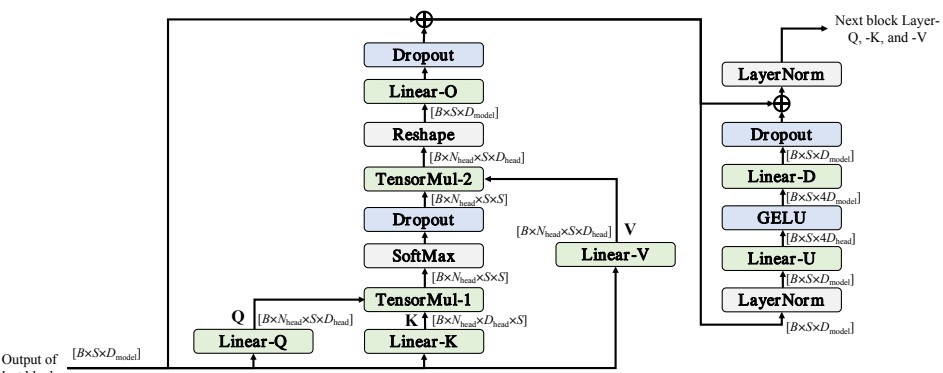

Fig. 5. The diagram of a single Transformer block. The shape of activations is annotated, where $B, S, D_{\text{model}}, N_{\text{head}}$, and $D_{\text{head}}$ are the batch size, sequence length, hidden size, number of attention heads, and head dimension, respectively. WTA-CRS can be applied to the operators in green; the activation maps of operators in blue can be losslessly compressed; and those in gray are not compressed in this paper. The idea of this figure is inspired by [32].

To ensure the unbiasness of gradient and reduce the memory usage of storing $\mathbf{H}$, as shown in the example of Figure 4, **we only replace** GEMM **in the backward pass with its approximation (e.g., Equation (1c)), while leaving the forward one unchanged (e.g., Equation (1a)). We show in Appendix B that the estimated weight gradient is unbiased in this case.**

**System Implementation.** Here we present how we implement WTA-CRS in Equation (6) in practice. For the linear layer, as we analyzed, we only replace GEMM in Equation (1c) with its approximated version. In this case, $\mathbf{X}$ and $\mathbf{Y}$ in Equation (6) are activation $\mathbf{H}^{\top}$ and output gradient $\nabla \mathbf{Z}$, respectively. Given the total column-row pair budget $k$, the **first** step is to build the deterministic index set $\mathcal{C}$, where each element is summed explicitly without sampling. Note that $\mathcal{C}$ is a set of indices with the highest probabilities in Equation (3). Thus, to build $\mathcal{C}$, we only need to determine its size, denoted as $|\mathcal{C}|$, which minimizes the variance of the estimator. As Theorem 2 suggested, we set $|\mathcal{C}| = \min_{|\mathcal{C}| \in \{0, \cdots, k\}} \frac{1 - \sum_{c \in \mathcal{C}} p_c}{k - |\mathcal{C}|}$. The **second** step is to sample $k - |\mathcal{C}|$ column-row indices from the remaining distribution $\mathcal{P}^{\mathcal{D} \setminus \mathcal{C}}$ to obtain the set $\mathcal{C}_{\text{stoc}}$, where $|\mathcal{C}_{\text{stoc}}| = k - |\mathcal{C}|$. The **third** step is to build sub-sampled $\mathbf{H}'$ with only rows from $\mathcal{C} \cup \mathcal{C}_{\text{stoc}}$. Note that for rows in $\mathbf{H}'$ from $\mathcal{C}_{\text{stoc}}$, we need to normalize it by $\frac{1 - \sum_{c \in \mathcal{C}} p_c}{k - |\mathcal{C}|}$ according to Equation (6). We illustrate the above process in Figure 4. The pseudocode to Appendix D Algorithm 1.

**Scope.** Here we show which operation can be replaced with its approximation version. As shown in Figure 5, the transformer is mainly consisted of linear layer, TensorMul, and other operations (e.g., GeLU, Dropout, LayerNorm). TensorMul in Figure 5 refers to the multiplication between two four-dimensional tensors. Our WTA-CRS *can be applied to Linear-Q, -K, -V, -O, -U, -D, TensorMul-1, and TensorMul-2 (in green).* The activations of Dropout and GELU operations (in blue) can be losslessly compressed. The Softmax and LayerNorm operators (in gray) remain unchanged.

## 4 Related Work and Discussion

Due to the page limit, we discuss the related work on approximated matrix multiplication and aproximation in LLM inference. Other related topics, e.g., parameter-efficient fine-tuning, activation quantization, and gradient checkpointing, can be found in Appendix A. We also discuss the limitation and potential negative social impact in Appendix A.

**Approximated Matrix Multiplication.** In the context of neural networks, approximated matrix multiplication methods can be broadly categorized into two main groups: (1) Butterfly-based methods [33, 34] replace dense weight matrices with butterfly matrices. We note that they focus on the weight matrix and are orthogonal to our research, as we concentrate on sub-sampling the activation matrix. (2) Column-row sampling (CRS) methods[22, 21, 31] select important rows and columns from the input matrices and perform the multiplication on the sampled matrix. Our work is closely aligned

with this second research line. [21, 31] share similarities with our research in terms of utilizing CRS for approximating matrix multiplication within neural networks. The main distinction lies in how to select the column-row pairs. Specifically, [21] deterministically selects column-row pairs without scaling, whereas our estimator divides the column-row pairs into a deterministic component and a stochastic component. As we analyzed, selecting column-row pairs deterministically is biased. Later we show that this approach may cause a significant accuracy drop (**"Deterministic" in Figure 12**).

**Approximation in Inference** There are two approximation techniques to reduce inference latency. (1) Sparse Modeling, which only involve a subset of weights during computation to reduce both computational and memory I/O demands [35, 36, 37]. (2) Quantization, which compresses the trained weight into lower numerical precision [38, 39, 40]. All these approximation techniques trade-off model quality in return for improved efficiency. Besides approximation technique itself, researchers also propose methods to recover the accuracy drop of compressed models [41, 42], e.g., by prompting compressed models. These methods can greatly improve the accuracy-efficiency trade-off of LLMs.

# 5 Experiments

In this section, we design experiments to answer the following research questions: **RQ1:** How effective is WTA-CRS in terms of accuracy with reduced memory usage? **RQ2:** How sensitive is WTA-CRS affected by its key hyper-parameters? **RQ3:** WTA-CRS contains two parts, i.e., the deterministic summation part and the statistical sampling part. Are they both necessary? **RQ4:** How is the fine-tuning speed affected by WTA-CRS ?

## 5.1 Experiment Setup

**Datasets and Evaluation Protocol.** Following most of the previous work, we adopt GLUE benchmark [43] to evaluate the effectiveness of different methods, including the CoLA, SST-2, MRPC, QQP, MNLI, QNLI, RTE, and STS-B datasets. To evaluate the memory usage, we report the peak GPU memory usage and compression rate during the fine-tuning process with Huggingface API [44].

**Compared Methods and Adopted Models.** We consider three methods to compare in this paper: Full fine-tuning (Full), LoRA [12], and Ladder Side-tuning (LST) [9]. Specifically, **Full** tunes all of the parameters in the model to provide an upper bound of accuracy; **LoRA** inserts trainable low-rank matrices into the model to parameterize the weights' changes; **LST** injects a trainable ladder side structure. Since WTA-CRS essentially replace the linear operation with approximated one, **we emphasize that our WTA-CRS is compatible with all these three compared methods, i.e., they can be combined together towards smaller memory usage.** For the backbone model, we follow the previous work [9, 14, 12] to adopt the OPT [45], Bert-Base [2], Bert-Large, T5-Base, T5-Large, and T5-3B [46] for evaluating the effectiveness of different methods.

**Hyperparameter Settings.** For WTA-CRS, it only has one hyperparameter $k$, which controls the column-row pair budget. We assign the same $k$ to all replaceable linear operations in the model. We consider the normalized column-row pair budget $k/|\mathcal{D}| \in \{0.3, 0.1\}$, which are denoted as WTA-CRS @0.3 and WTA-CRS@0.1, respectively. We also consider the combination of WTA-CRS and LoRA to further reduce the memory cost of both optimizer and activations. The detailed hyperparameters are given in Appendix F. All reported results are averaged over three random trials.

## 5.2 Accuracy versus Memory Usage (RQ1)

To answer **RQ1**, we first analyze the trade-off between the model performance and memory saving. The evaluation results and peak memory usage are given in Tables 1 and 2, respectively. We observe:

❶ *WTA-CRS achieves a superior trade-off between accuracy and memory usage compared to baselines. Specifically, WTA-CRS has negligible accuracy drop, while the peak memory usage is reduced by* $2.1\times \sim 2.7\times$ (when combined with LoRA).

Table 1: The GLUE benchmark results with T5 and Bert at different scales.

| Model | Method | CoLA | SST-2 | MRPC | QQP | MNLI | QNLI | RTE | STS-B | AVG |
|---|---|---|---|---|---|---|---|---|---|---|
| BERT-Base | Full | 60.9±1.89 | 92.2±0.34 | 87.9±0.46 | 87.8±0.01 | 83.7±0.05 | 90.7±0.14 | 66.4±0.36 | 88.1±0.27 | 82.2 |
| | LoRA | 61.6±0.25 | 91.7±0.17 | 90.0±0.34 | 86.9±0.1 | 83.6±0.02 | 90.8±0.17 | 68.2±0.36 | 87.6±0.52 | 82.6 |
| | WTA-CRS@0.3 | 60.7±0.89 | 90.2±0.06 | 87.0±0.14 | 87.5±0.03 | 83.4±0.06 | 90.4±0.07 | 65.9±0.18 | 89.3±0.2 | 81.8 |
| | LoRA+WTA-CRS@0.3 | 61.5±1.08 | 89.6±0.52 | 89.6±0.09 | 86.3±0.02 | 82.8±0.35 | 90.6±0.16 | 67.9±0.72 | 87.3±0.7 | 81.9 |
| T5-Base | Full | 60.1±0.37 | 94.9±0.29 | 91.5±0.29 | 88.5±0.07 | 87.0±0.1 | 93.3±0.03 | 79.4±0.78 | 90.6±0.14 | 85.7 |
| | LoRA | 60.6±0.94 | 94.6±0.05 | 92.2±0.31 | 87.4±0.06 | 86.2±0.06 | 93.4±0.03 | 80.6±0.74 | 90.7±0.05 | 85.7 |
| | LST | 55.5±0.24 | 94.0±0.17 | 91.1±0.18 | 87.4±0.01 | 85.7±0.13 | 93.4±0.0 | 72.7±0.54 | 90.4±0.06 | 83.8 |
| | WTA-CRS@0.3 | 60.9±0.52 | 94.8±0.14 | 91.1±0.35 | 88.0±0.11 | 86.3±0.02 | 93.1±0.07 | 78.7±0.59 | 90.5±0.05 | 85.4 |
| | LoRA+WTA-CRS@0.3 | 60.0±0.51 | 94.4±0.16 | 92.0±0.38 | 87.3±0.04 | 85.6±0.08 | 93.2±0.01 | 80.1±1.02 | 90.4±0.06 | 85.4 |
| BERT-Large | Full | 66.8±0.31 | 93.5±0.29 | 89.5±0.26 | 88.5±0.03 | 86.4±0.19 | 92.1±0.24 | 72.6±0.36 | 90.2±0.76 | 85.0 |
| | LoRA | 65.9±0.27 | 93.8±0.17 | 90.8±0.37 | 87.6±0.08 | 85.9±0.05 | 92.0±0.2 | 71.3±0.18 | 90.3±0.09 | 84.7 |
| | WTA-CRS@0.3 | 64.7±0.44 | 93.5±0.0 | 89.3±0.39 | 88.2±0.04 | 85.2±0.03 | 91.9±0.12 | 73.8±0.54 | 90.4±0.02 | 84.6 |
| | LoRA+WTA-CRS@0.3 | 66.0±0.33 | 93.3±0.29 | 89.7±1.32 | 87.6±0.02 | 86.0±0.07 | 91.9±0.14 | 72.4±0.17 | 89.7±0.04 | 84.6 |
| T5-Large | Full | 61.3±1.01 | 96.3±0.0 | 93.4±0.13 | 89.7±0.01 | 89.8±0.07 | 94.2±0.05 | 85.3±0.17 | 91.8±0.08 | 87.7 |
| | LoRA | 63.3±0.26 | 96.4±0.14 | 93.5±0.16 | 88.5±0.03 | 89.5±0.05 | 94.3±0.07 | 84.2±0.68 | 91.7±0.13 | 87.7 |
| | LST | 59.9±0.77 | 95.8±0.06 | 91.8±0.08 | 88.4±0.01 | 88.7±0.05 | 94.2±0.02 | 82.5±0.18 | 91.4±0.07 | 86.6 |
| | WTA-CRS@0.3 | 60.9±1.18 | 96.3±0.25 | 93.6±0.57 | 89.3±0.04 | 89.5±0.12 | 94.1±0.03 | 84.4±0.34 | 91.3±0.05 | 87.4 |
| | LoRA+WTA-CRS@0.3 | 62.9±1.19 | 96.2±0.05 | 93.6±0.47 | 88.3±0.02 | 89.2±0.08 | 94.0±0.07 | 83.9±0.95 | 91.3±0.03 | 87.4 |
| T5-3B | LoRA | 70.1±0.37 | 96.8±0.29 | 94.0±0.27 | 89.9±0.0 | 91.0±0.14 | 95.6±0.05 | 85.9±0.36 | 92.9±0.08 | 89.5 |
| | LoRA+WTA-CRS@0.3 | 71.4±0.35 | 96.4±0.06 | 94.6±0.39 | 90.0±0.05 | 91.0±0.06 | 95.6±0.12 | 86.3±0.36 | 92.9±0.09 | 89.8 |

Table 2: Peak memory usage (GB) and compression rate of fine-tuning T5-Base and -Large. We measure the memory usage on a single NVIDIA A100 (80GB) GPU. For T5-3B, since it is trained using multi-GPUs with data parallel. We instead report the maximum batch size in Figure 6 for it.

| | FP | LoRA | LST | WTA-CRS@0.3 | WTA-CRS@0.1 | LoRA+WTA-CRS@0.3 | LoRA+WTA-CRS@0.1 |
|---|---|---|---|---|---|---|---|
| T5-Base | 17.66 (1×) | 13.84 (1.3×) | 5.50 (3.2×) | 8.44 (2.1×) | 7.30 (2.4×) | 6.50 (2.7×) | 5.44 (3.2×) |
| T5-Large | 45.85 (1×) | 36.83 (1.2×) | 14.85 (3.1×) | 21.58 (2.1×) | 18.46 (2.5×) | 17.44 (2.6×) | 14.64 (3.13×) |

As we analyzed, LoRA mainly reduces the memory of optimizer states. Thus, although it has negligible accuracy drop, it can only achieve $\sim 1.3\times$ peak memory saving. LST can reduce memory usage up to $3\times$, but its accuracy drop is much larger than LoRA and WTA-CRS. Since WTA-CRS executes at the operation level and focuses on the activations, we further combine LoRA with WTA-CRS to reduce memory usage more aggressively. When combing with LoRA, WTA-CRS achieves $2.7\times$ memory usage saving with almost no accuracy drop. To fully tune T5-3B, it requires 66.4GB of memory for in a mini-batch size of 32, which relies on A100 GPUs with a capacity of 80GB. On the other hand, LoRA+WTA-CRS only requires 21.6GB of memory for finetuning with a mini-batch size of 32, which can run on a GPU with 24GB memory, e.g. RTX3090Ti or A5000. We have experimentally confirmed this

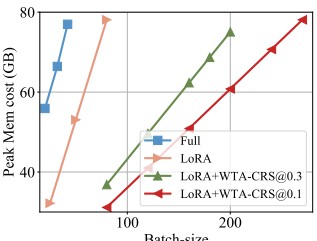

Fig. 6. Peak memory usage versus the maximum batch size of T5-3B. More similar results are shown in Appendix E.2.

conclusion. Under the same hardware, WTA-CRS enables the tuning of larger models, resulting in improved down-streaming task performance. Thus as shown in Figure 1, ❷ *under the similar memory budget, WTA-CRS outperforms the other methods in terms of the accuracy.*

Also, according to Figure 6, for T5-3B, LoRA itself can enable $1.9\times$ larger batch size. ❸ *When combined with LoRA, WTA-CRS enables $4.8\times$ ($k=0.3|\mathcal{D}|$) to $6.4\times$ ($k=0.1|\mathcal{D}|$) larger batch-size.*

**Influence of Row-column Pairs Budget (RQ2).** As we analyzed in Section 3.2, WTA-CRS only have one hyperparameter, i.e., the total column-row pair budgets $k$. We conduct the ablation study with different budget $k$ in Figure 7. We observe that ❹ *It has almost no accuracy drop when $k = 0.3|\mathcal{D}|$. And the accuracy drop is about $1\%$ when $k = 0.1|\mathcal{D}|$. Notably, for T5-3B, the accuracy drop is only $0.4\%$ when $k = 0.1|\mathcal{D}|$, which is much smaller than T5-Base and T5-Large.* This suggests that larger models are more compressible because they have more redundant activations, which is consistent with previous observations [47].

### 5.3 Ablation Study (RQ3 and RQ4)

To answer **RQ3**, WTA-CRS is compared with two compositional methods to demonstrate its superiority. Namely, (1) the **Deterministic** method selects row-column pairs with top $k$ probability of Equation (3). We note that this is the estimator proposed in [21]. (2) The **CRS** method follows Equation (3) to

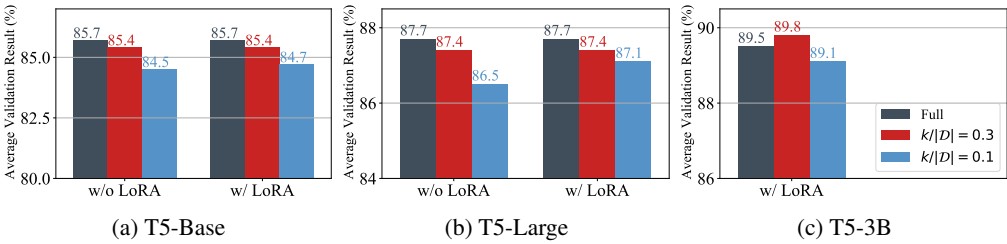

Fig. 7. Average validation results on GLUE dataset of `WTA-CRS` with varying budgets.

sample the row-column pairs. All methods are deployed to `GEMM` in the backward pass, while leaving the forward one unchanged. The experiments are conducted on the training of T5-base language model on the SST2, MNLI, and QQP datasets; The column-row pair budget takes $k/|\mathcal{D}| = 0.1$ for all methods. The validation accuracy versus training epoch is given in Figure 12. We observe:

❺ *`WTA-CRS` outperforms all compared methods, especially as the training epoch grows.* The deterministic selection of top $k$ column-row pairs suffers from accumulation of bias error that ultimately results in a failure of convergence. For CRS, it also enables the unbiased weight gradient. However, as we theoretically and experimentally analyzed in Theorem 7 and Figure 3, it is worse than `WTA-CRS` due to larger variance. In summary, both the deterministic and stochastic parts contribute to the effectiveness of `WTA-CRS` , which is consistent with our theoretical analysis.

**The speed of `WTA-CRS` (RQ4).** The configuration of computational infrastructure is given in Appendix F.1. We note that `WTA-CRS` does not add any extra parameters to the model. Thus, `WTA-CRS` only affect the fine-tuning speed, without affecting the inference speed. Below we analyze how the fine-tuning speed affected by `WTA-CRS` . As we analyzed in Appendix A limitation, the current implementation is not heavily optimized and thus the execution time of `WTA-CRS` is still slower than the original linear operation (details are shown in Appendix E.2). However, under the same

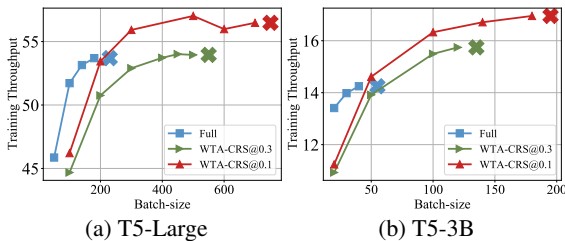

Fig. 8. Batch size versus training throughput (sentences/sec) with different methods, where the sequential length is 128. The hardware is one single NVIDIA-A100 (80GB).

hardware, a reduction in activation memory enables the use of larger batch sizes, thereby improving training speed due to increased GPU utilization [48, 9]. As we analyzed in Figure 6, `WTA-CRS` can enlarge the available batch size by up to $4.8\times$ larger. This enhancement is expected to result in a acceleration of the training speed. To illustrate, Figure 8 presents a visualization of batch size against training throughput (sentences per second) for both T5-Large and T5-3B models. We observe that
❻ *`WTA-CRS` enables faster training speed under the same hardware.* Specifically, on the T5-Large model, `WTA-CRS`@0.1 shows $1.08\times$ higher training throughput; and on the T5-3B model, `WTA-CRS`@0.3 and `WTA-CRS`@0.1 achieve $1.14\times$ and $1.21\times$ higher training throughput, respectively.

## 6 Acknowledgements

The authors thank the anonymous reviewers for their helpful comments. The work is in part supported by NSF grants NSF IIS-1849085, IIS-2224843, and NSF Awards 2117439 and 2112606. This work made use of the High Performance Computing Resource in the Core Facility for Advanced Research Computing at Case Western Reserve University.

## 7 Conclusion

In this paper, we propose `WTA-CRS` , a new unbiased estimator for matrix production with reduced variance. We theoretically and experimentally show when and why the estimator is better than the traditional unbiased estimator in terms of the variance. In the context of adapting transformers, it almost has no accuracy drop while reducing the peak memory usage by up to $2.7\times$, and it enables a $6.4\times$ larger batch size, which in return resulting in $1.2\times$ higher training throughput.

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

# Appendix

## A  Extended Related Work and Discussion

**Parameter-Efficient Fine-tuning.**  Parameter-efficient tuning methods select a small subset of parameters or insert a few parameters to a pre-trained network. Then they only update the small subset of parameters, while keeping others fixed [8, 9, 10, 11, 12, 13, 14, 49]. For example, Adapters [14, 13] insert a small module into the transformer blocks and only update it. Similarly, prompt tuning [8] introduces a small vector that is concatenated with the input embeddings. BitFit [11] only tunes the bias term of the model. LoRA [12] injects trainable rank decomposition matrices into the transformer block. Although these methods are "parameter-efficient", they actually cannot reduce the memory usage of the model itself. This is because we still needs to build the computation graph for the whole model. Instead, the memory usage of optimizer states will be significantly reduced, which is in proportional to the number of trainable parameters [15].

**Activation Quantization.**  The activation quantization methods focus on quantizing the activation into low numerical precision numbers, e.g., 8-bit integers [16, 30, 50, 51, 52]. Our work also compresses the activation, but in a different way. *We emphasize that our work is orthogonal to activation quantization in the sense that our work essentially reduces the dimension of activation.* This distinction allows our method to be readily combined with activation quantization techniques, offering the potential for even more aggressive compression.

**Gradient Checkpointing.**  Gradient checkpointing helps decrease activation memory usage by saving only a selection of activations. However, it demands additional computation during the backward pass, as discarded activations must be recalculated [18, 17]. According to the report of Checkmate[3] [17], it achieves "a 2.3x memory reduction when training a BERT model with Checkmate optimizations (at 1x extra overhead for rematerialization)".

**Limitations**  Although `WTA-CRS` significantly reduces the computation of the backward pass in a hardware-friendly way i.e., dropping entire rows/columns in the tensor, the current implementation still hampers the execution time of linear operations. This is because the extra sampling process and data movement counteract the acceleration. However, we note that (1) the overhead can be greatly reduced with better implementation, e.g., using prefetch and operation-fusion technique [30]; (2) the existing implementation can still yield a large speedup when employing larger batch sizes (Figure 8).

**Potential Negative Societal Impacts.**  Our research primarily focuses on reducing the memory requirement of fine-tuning Language Models (LMs). The carbon emissions produced by LM fine-tuning may pose environmental issues. Our next step is to further improve the efficiency of LM fine-tuning, particularly on hardware with lower energy consumption.

## B  Unbiasedness of Weight Gradient

This part we directly follow the proof of Theorem 1 in ActNN [16]. For completeness, we provide the proof sketch here that is short and easy to follow. Specifically, here we use ReLU as the activation function for illustration convenience. We note that the conclusion in this section holds for any non-linear activation function. Specifically, the forward pass of ReLU-Linear at the $l^{\text{th}}$ layer is

$$\mathbf{Z}^{(l+1)} = \mathbf{H}^{(l)}\mathbf{W}^{(l)},$$
$$\mathbf{H}^{(l+1)} = \text{ReLU}(\mathbf{Z}^{(l+1)}),$$

and the backward pass of ReLU is:

$$\mathbb{E}[\nabla\mathbf{Z}^{(l+1)}] = \mathbb{E}[\mathbb{1}_{\mathbf{Z}^{(l+1)}>0} \odot \nabla\mathbf{H}^{(l+1)}]$$
$$= \mathbb{1}_{\mathbf{Z}^{(l+1)}>0} \odot \mathbb{E}[\nabla\mathbf{H}^{(l+1)}],$$

---

[3]`https://github.com/parasj/checkmate/issues/153`

where $\odot$ is the element-wise product and $\mathbb{1}$ is the indicator function. The element-wise product is linear operation and $\mathbb{1}_{\mathbf{Z}^{(l+1)}>0}$ is only related to the pre-activation $\mathbf{Z}^{(l+1)}$ in the forward pass. We only apply the approximation during the backward pass so $\mathbb{1}_{\mathbf{Z}^{(l+1)}>0}$ can be extracted from the expectation. We know that for the last layer $L$, we have $\mathbb{E}[\nabla\mathbf{H}^{(L)}] = \mathbf{H}^{(L)}$ since we do not apply activation at the output layer. We then can prove by induction that $\mathbb{E}[\nabla\mathbf{H}^{(l+1)}] = \mathbf{H}^{(l+1)}$ and $\mathbb{E}[\nabla\mathbf{W}^{(l)}] = \mathbf{W}^{(l)}$ for any layer $l$.

## C   Proof

### C.1   Derivation of Equation (3)

Let $\mathbf{X} \in \mathbb{R}^{n\times m}$, $\mathbf{Y} \in \mathbb{R}^{m\times q}$ be two matrices. The matrix multiplication $\mathbf{XY}$ can be estimated as

$$\texttt{GEMM}(\mathbf{X}, \mathbf{Y}) = \sum_{i=1}^{m} \mathbf{X}_{:,i}\mathbf{Y}_{i,:} \approx \sum_{t=1}^{k} \frac{1}{kp_{i_t}}\mathbf{X}_{:,i_t}\mathbf{Y}_{i_t,:} = \mathbf{X}'\mathbf{Y}',$$

Equation (3) shows the approximation error $\mathbb{E}[||\mathbf{XY} - \mathbf{X}'\mathbf{Y}'||_F]$ is minimized when the probabilities

$$p_i = \frac{||\mathbf{X}_{:,i}||_2\,||\mathbf{Y}_{i,:}||_2}{\sum_{j=1}^{m} ||\mathbf{X}_{:,j}||_2\,||\mathbf{Y}_{j,:}||_2}.$$

*Proof.* Let $f(i) = \frac{\mathbf{X}_{:i}\mathbf{Y}_{i:}}{p_i} \in \mathbb{R}^{n\times q}$. We note that $f(i)$ is an unbiased estimation of $\mathbf{XY}$. Namely,

$$\mathbb{E}_{j\sim\mathcal{P}}[f(j)] = \sum_{i=1}^{m} p_i\frac{\mathbf{X}_{:,i}\mathbf{Y}_{i:}}{p_i} = \mathbf{XY}.$$

Then we have

$$\mathbf{X}'\mathbf{Y}' = \frac{1}{k}\sum_{t=1}^{k} f(i_t), \tag{8}$$

where $i_1, \cdots, i_t$ are the index of the sampled column-row pairs at $t^{\text{th}}$ random trials. For each $i_t$, its variance is

$$\begin{aligned}
\mathrm{Var}[f(i_t)] &= \mathrm{Var}[\frac{\mathbf{X}_{:i_t}\mathbf{Y}_{i_t:}}{p_{i_t}}] \\
&= \mathbb{E}[\frac{\mathbf{X}_{:i_t}^2\mathbf{Y}_{i_t:}^2}{p_{i_t}^2}] - \mathbb{E}^2[\frac{\mathbf{X}_{:i_t}\mathbf{Y}_{i_t:}}{p_{i_t}}] \\
&= \mathbb{E}[\frac{\mathbf{X}_{:i_t}^2\mathbf{Y}_{i_t:}^2}{p_{i_t}^2}] - (\mathbf{XY})^2. \\
&= \sum_{t=1}^{m} \frac{\mathbf{X}_{:t}^2\mathbf{Y}_{t:}^2}{p_t} - (\mathbf{XY})^2. \tag{9}
\end{aligned}$$

where the first step follows from the fact that $\mathrm{Var}[\mathbf{x}] = \mathbb{E}[\mathbf{x}^2] - \mathbb{E}^2[\mathbf{x}]$.

Then we have,

$$\begin{aligned}
\mathbb{E}[||\mathbf{XY} - \mathbf{X}'\mathbf{Y}'||_F] &= \sum_{i=1}^{n}\sum_{j=1}^{q} \mathbb{E}[(\mathbf{XY} - \mathbf{X}'\mathbf{Y}')_{ij}^2] \\
&= \sum_{i=1}^{n}\sum_{j=1}^{q} \mathrm{Var}[(\mathbf{X}'\mathbf{Y}')_{ij}].
\end{aligned}$$

By combining Equation (8) and Equation (9) into the above equation, we have

$$\mathbb{E}[||\mathbf{XY} - \mathbf{X}'\mathbf{Y}'||_F] = \frac{1}{k}\sum_{i=1}^{n}\sum_{j=1}^{q}\sum_{t=1}^{m}\frac{\mathbf{X}_{it}^2\mathbf{Y}_{tj}^2}{p_t} - \frac{1}{k}||\mathbf{XY}||_F^2.$$

$$= \frac{1}{k}\sum_{t=1}^{m}\frac{||\mathbf{X}_{:,t}||_2^2||\mathbf{Y}_{t,:}||_2^2}{p_t} - \frac{1}{k}||\mathbf{XY}||_F^2.$$

To minimize $\mathbb{E}[||\mathbf{XY} - \mathbf{X}'\mathbf{Y}'||_F]$, the optimal probability distribution can be obtained via solving the following optimization problem:

$$\min_{p_1,\cdots,p_m}\sum_{t=1}^{m}\frac{||\mathbf{X}_{:,t}||_2^2||\mathbf{Y}_{t,:}||_2^2}{p_t},$$

$$\text{s.t.}\sum_{t=1}^{m}p_t = 1.$$

The solution to the above convex problem is the distribution defined in Equation (3). Namely,

$$p_i = \frac{||\mathbf{X}_{:,i}||_2\,||\mathbf{Y}_{i,:}||_2}{\sum_{j=1}^{m}||\mathbf{X}_{:,j}||_2\,||\mathbf{Y}_{j,:}||_2}.$$

$\square$

## C.2 Unbiasedness of Our Proposed Estimator

**Theorem 1** (Proof in Appendix C.2). *The estimator defined in Equation (4) is an unbiased estimator for matrix production* $\mathbf{XY}$*, i.e,* $\mathbb{E}_{j\sim\mathcal{P}^{\mathcal{D}\backslash\mathcal{C}}}[\sum_{c\in\mathcal{C}}f(c)p_c + (1 - \sum_{c\in\mathcal{C}}p_c)f(j)] = \mathbf{XY}.$

*Proof.*

$$\mathbb{E}_{j\sim\mathcal{P}^{\mathcal{D}\backslash\mathcal{C}}}\Big[\sum_{c\in\mathcal{C}}f(c)p_c + (1 - \sum_{c\in\mathcal{C}}p_c)f(j)\Big]$$

$$= \sum_{c\in\mathcal{C}}f(c)p_c + (1 - \sum_{c\in\mathcal{C}}p_c)\mathbb{E}_{j\sim\mathcal{P}^{\mathcal{D}\backslash\mathcal{C}}}[f(j)]$$

$$= \sum_{c\in\mathcal{C}}f(c)p_c + (1 - \sum_{c\in\mathcal{C}}p_c)\sum_{j\in\mathcal{D}\backslash\mathcal{C}}\frac{p_j}{1 - \sum_{c\in\mathcal{C}}p_c}f(j)$$

$$= \sum_{c\in\mathcal{C}}f(c)p_c + \sum_{j\in\mathcal{D}\backslash\mathcal{C}}f(j)p_j$$

$$= \mathbb{E}_{j\sim\mathcal{P}}[f(j)]$$

$$= \mathbf{XY}$$

$\square$

## C.3 Variance of Our Proposed Estimator

**Theorem 2** (Proof in Appendix C.3). *Suppose the total budget of column-row pairs is $k$. If $\mathcal{C}$ satisfies*

$$\sum_{c\in\mathcal{C}}p_c > \frac{|\mathcal{C}|}{k},\tag{7}$$

*then we have* $\text{Var}[\hat{g}(\mathbf{X},\mathbf{Y})] < \text{Var}[g(\mathbf{X},\mathbf{Y})]$*. Moreover,* $\text{Var}[\hat{g}(\mathbf{X},\mathbf{Y})]$ *is minimized when* $|\mathcal{C}| = \min_{|\mathcal{C}|\in\{0,\cdots,k\}}\frac{1 - \sum_{c\in\mathcal{C}}p_c}{k - |\mathcal{C}|}.$

*Proof.* Recall that the original estimator for matrix production $\mathbf{X}\mathbf{Y}$ is defined as

$$\mathbb{E}_{i\sim\mathcal{P}}[f(i)]. \tag{10}$$

and our proposed family of estimator is defined as:

$$h(j) = \mathbb{E}_{j\sim\mathcal{P}^{\mathcal{D}\backslash\mathcal{C}}}\Big[\sum_{c\in\mathcal{C}} f(c)p_c + (1 - \sum_{c\in\mathcal{C}} p_c)f(j)\Big]. \tag{11}$$

We first define three independent random variables as belows:

$$u \sim \mathcal{P}^{\mathcal{C}}, \tag{12}$$

$$j \sim \mathcal{P}^{\mathcal{D}\backslash\mathcal{C}}, \tag{13}$$

$$b \sim \text{Bernoulli}(1 - \sum_{c\in\mathcal{C}} p_c). \tag{14}$$

According to the Law of total variance, we have

$$\begin{aligned}
\text{Var}[f(i)] &= \mathbb{E}_b\Big[Var[f(i)|b]\Big] + \text{Var}_b\Big[\mathbb{E}[f(i)|b]\Big] \\
&\geq \mathbb{E}_b\Big[\text{Var}[f(i)|b]\Big] \\
&= \sum_{c\in\mathcal{C}} p_c\text{Var}[f(i)|b = 0] + (1 - \sum_{c\in\mathcal{C}} p_c)\text{Var}[f(i)|b = 1] \\
&\geq (1 - \sum_{c\in\mathcal{C}} p_c)\text{Var}[f(i)|i \in \mathcal{D}\backslash\mathcal{C}] \tag{15}
\end{aligned}$$

where the first step follows from the fact that for any random variance $\mathbf{x}, \mathbf{y}$, we have $\text{Var}[\mathbf{y}] = \mathbb{E}[\text{Var}[\mathbf{y}|\mathbf{x}]] + \text{Var}[\mathbb{E}[\mathbf{y}|\mathbf{x}]]$. Also, by Equation (11), we have

$$\text{Var}[h(j)] = (1 - \sum_{c\in\mathcal{C}} p_c)^2\text{Var}[f(j)|j \in \mathcal{D}\backslash\mathcal{C}]. \tag{16}$$

By combining the above two inequality, we have

$$\text{Var}[h(j)] \leq (1 - \sum_{c\in\mathcal{C}} p_c)\text{Var}[f(i)]. \tag{17}$$

Equation (17) quantitatively shows the variance reduction of $h(j)$ over $f(i)$. Then we compare our estimator $\hat{g}(\mathbf{X}, \mathbf{Y})$ and $g(\mathbf{X}, \mathbf{Y})$ in terms of variance.

First, because $g(\mathbf{X}, \mathbf{Y}) = \frac{1}{k}\sum_{t=1}^k f(i_t)$, $i_1, \cdots i_k \overset{\text{i.i.d}}{\sim} \mathcal{P}$. Thus we have

$$\text{Var}[g(\mathbf{X}, \mathbf{Y})] = \frac{1}{k}\text{Var}[f(i)]. \tag{18}$$

Similarly, we have

$$\text{Var}[\hat{g}(\mathbf{X}, \mathbf{Y})] = \frac{1}{k - |\mathcal{C}|}\text{Var}[h(j)]. \tag{19}$$

By combining Equation (17) into the above two equations, we have

$$\begin{aligned}
\text{Var}[\hat{g}(\mathbf{X}, \mathbf{Y})] &= \frac{1}{k - |\mathcal{C}|}\text{Var}[h(j)] \tag{20} \\
&\leq \frac{1 - \sum_{c\in\mathcal{C}} p_c}{k - |\mathcal{C}|}\text{Var}[f(i)] \\
&\leq \frac{1 - \sum_{c\in\mathcal{C}} p_c}{k - |\mathcal{C}|}k\text{Var}[g(\mathbf{X}, \mathbf{Y})],
\end{aligned}$$

where the first step follows from Equation (17). By setting $\frac{1-\sum_{c\in\mathcal{C}} p_c}{k-|\mathcal{C}|}k \le 1$, we arrive the conclusion that when $\sum_{c\in\mathcal{C}} p_c > \frac{|\mathcal{C}|}{k}$, we have $\mathrm{Var}[\hat{g}(\mathbf{X},\mathbf{Y})] \le \mathrm{Var}[g(\mathbf{X},\mathbf{Y})]$.

Further, $\frac{1-\sum_{c\in\mathcal{C}} p_c}{k-|\mathcal{C}|}k$ achieves the minimal when $|\mathcal{C}| = \min_{|\mathcal{C}|\in\{0,\cdots,k\}} \frac{1-\sum_{c\in\mathcal{C}} p_c}{k-|\mathcal{C}|}$.

$\square$

## D   Implementation Details

The pseudocode for approximated linear layer with `WTA-CRS` and standard line layer is given in Algorithm 1 and Algorithm 3, respectively. The column-row pair sampling procedure is given in Algorithm 2. For the ease of illustration, we ignore the sequential length. As we mentioned in the main text, we only replace the `GEMM` in the backward pass with `WTA-CRS` . According to Equation (1c), we need the activation gradient $\nabla\mathbf{Z}$ to perform the column-row pair sampling during the forward pass. Thus we initialize a cache in CPU memory to store the gradient norm of activations from the last step. When performing column-row pair selection, we need to swap the gradient norm of activations between CPU and GPU, which will cause extra time overhead due to the data movement. Fortunately, we note that the number of elements in the gradient norm of activations is significantly less than the one in activations, which does not cause a significant time overhead.

---

**Algorithm 1:** Forward & Backward pass of Approximated Linear Layer

---

**Hyperparameter:** The total budget of column-row pairs $k$.
**procedure** INIT**:**
    Initialize `Cache` $\in \mathbb{R}^N$ as an empty matrix in main memory // $N$ `is the total number`
        `of samples in the dataset.  Cache is used for saving the norm of`
        `output gradient` $\nabla\mathbf{Z}$.
**end**
**procedure** FORWARD PASS**:**
    **Input:** activation $\mathbf{H} \in \mathbb{R}^{B\times D}$, weight $\mathbf{W} \in \mathbb{R}^{D\times D}$, indices of the current batch samples
        $BI = \{j_1,\cdots,j_B\}$.
    `ctx` $\leftarrow \{\}$ // `the context which saves tensors for backward`
    $\mathbf{Z} = \mathbf{HW}$
    $\mathbf{H}', ind \leftarrow$ SUBSAMPLE$(\mathbf{H}, \texttt{Cache}[BI], k)$
    // `Cache`$[BI]$ `is the cached gradient norm from the backward pass;` $ind$
        `is the set of involved column-row pair indices`
    `ctx` $\leftarrow \{\mathbf{H}', \mathbf{W}, BI, ind\}$
    **return** $\mathbf{Z}$
**end**
**procedure** BACKWARD PASS**:**
    **Input:** `ctx` from the forward pass, output gradient $\nabla\mathbf{Z} \in \mathbb{R}^{B\times D}$
    $\mathbf{H}', \mathbf{W}, BI, ind \leftarrow$ `ctx`
    $\nabla\mathbf{H} = \nabla\mathbf{Z}\mathbf{W}^\top$
    $\nabla\mathbf{Z}' \leftarrow \nabla\mathbf{Z}[ind]$
    // $\nabla\mathbf{Z}' \in \mathbb{R}^{k\times D}$
    $\nabla\mathbf{W} = \mathbf{H}'^\top\nabla\mathbf{Z}'$
    **for** $j$ in $BI$ **do**
        `Cache`$[j] = \|\nabla\mathbf{Z}_{j,:}\|_2$
    **end**
    // `Update the gradient norm of samples in the current batch`
    **return** $\nabla\mathbf{H}, \nabla\mathbf{W}$
**end**

---

---

**Algorithm 2:** SUBSAMPLE

---

**Input:** activation $\mathbf{H} \in \mathbb{R}^{B \times D}$, gradient norm $\mathbf{z} \in \mathbb{R}^B$, the total budget of column-row pairs $k$.

**for** $i = 1, \cdots, B$ **do**

$\quad p_i \leftarrow \frac{\mathbf{z}_i \|\mathbf{H}_{i,:}\|_2}{\sum_{j=1}^B \mathbf{z}_i \|\mathbf{H}_{j,:}\|_2}$ // The probability of column-row pairs defined in Equation (3).

**end**

$\hat{k} \leftarrow \min_{\hat{k} \in \{0, \cdots, k\}} \frac{1 - \sum_{c \in \mathcal{C}} p_c}{k - \hat{k}}$, s.t. $\mathcal{C} = |\hat{k}|$. // $\mathcal{C}$ is the set of column-row pair indices associated with $|\mathcal{C}|$ largest $p_i$.

Sample $k - |\mathcal{C}|$ i.i.d. column-row pairs $\mathcal{C}_{\text{stoc}} = \{i_1, \cdots, i_{k-|\mathcal{C}|}\}$ from the distribution $\mathcal{P}^{\mathcal{D} \setminus \mathcal{C}}$

$ind \leftarrow \mathcal{C} \cup \mathcal{C}_{\text{stoc}}$

**for** $j \in \mathcal{C}_{stoc}$ **do**

$\quad \mathbf{H}[j,:] \leftarrow \mathbf{H}[j,:] * \frac{1 - \sum_{c \in \mathcal{C}} p_c}{(k - |\mathcal{C}|) p_j}$  // We need to normalize the stochastic part in Equation (6) to ensure the unbiasedness.

**end**

$\mathbf{H}' \leftarrow \mathbf{H}[ind]$  //  $\mathbf{H}' \in \mathbb{R}^{k \times D}$

**return** $\mathbf{H}'$, *ind*

---

---

**Algorithm 3:** Forward & Backward pass of the standard Linear layer

---

**procedure** FORWARD PASS**:**

$\quad$ **Input:** activation $\mathbf{H}_Q \in \mathbb{R}^{BS \times D}$, weight $\mathbf{W}_Q \in \mathbb{R}^{D \times D}$, batch indices $index$

$\quad$ ctx $\leftarrow \{\}$ // the context which saves tensors for backward

$\quad \mathbf{Z}_Q = \mathbf{H}_Q \mathbf{W}_Q$

$\quad$ ctx $\leftarrow \{\mathbf{H}_Q, \mathbf{W}_Q\}$

$\quad$ **return** $\mathbf{Z}_Q$

**end**

**procedure** BACKWARD PASS**:**

$\quad$ **Input:** ctx from the forward pass, output gradient $\nabla \mathbf{Z}_Q$

$\quad \mathbf{H}_Q, \mathbf{W}_Q \leftarrow$ ctx

$\quad \nabla \mathbf{H}_Q = \nabla \mathbf{Z}_Q \mathbf{W}_Q^\top$

$\quad \nabla \mathbf{W}_Q = \mathbf{H}_Q^\top \nabla \mathbf{Z}_Q$

$\quad$ **return** $\nabla \mathbf{H}_Q, \nabla \mathbf{W}_Q$

**end**

---

# E   More Experimental Results

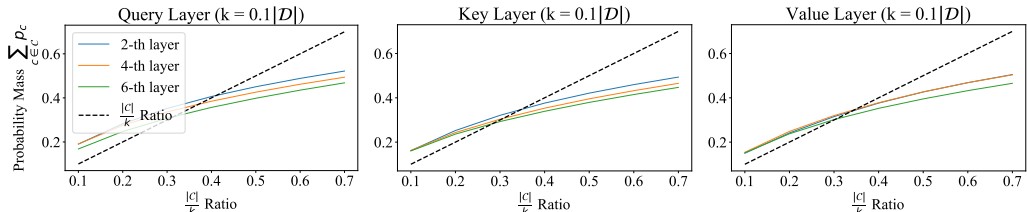

Fig. 9. The probability mass $\sum_{c \in \mathcal{C}} p_c$ versus $\frac{|\mathcal{C}|}{k}$ in Equation (7) at $k = 0.1|\mathcal{D}|$. Here we visualize the column-row index distribution of query/key/value layer T5-base model, fine-tuned on RTE dataset.

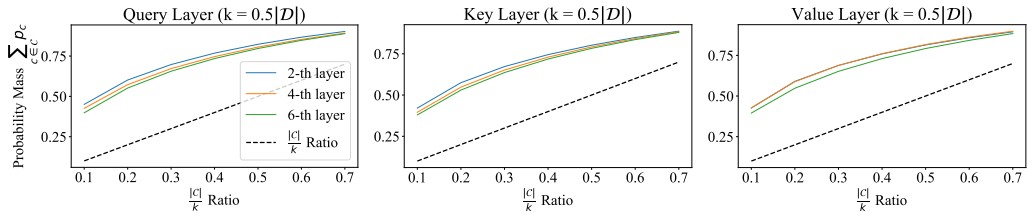

Fig. 10. The probability mass $\sum_{c \in \mathcal{C}} p_c$ versus $\frac{|\mathcal{C}|}{k}$ in Equation (7) at $k = 0.5|\mathcal{D}|$. Here we visualize the column-row index distribution of query/key/value layer T5-base model, fine-tuned on RTE dataset.

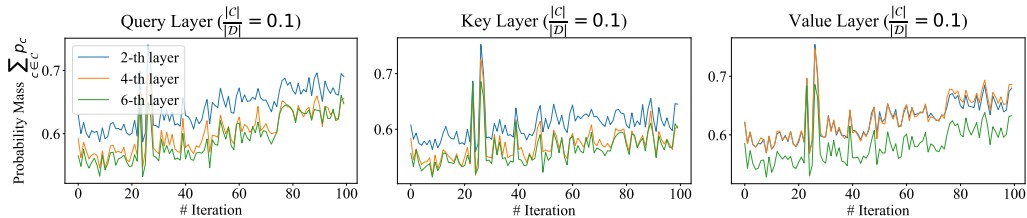

Fig. 11. The probability mass of top-$10\%$ column-row pairs in Equation (3) versus iterations. Here we visualize the query/key/value layer T5-base model, fine-tuned on RTE dataset.

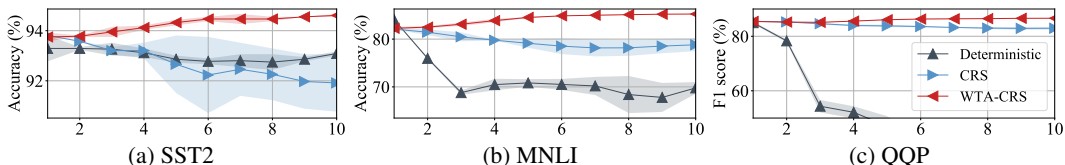

Fig. 12. Validation results of T5-Base with different methods.

## E.1   More Experimental Analysis on Theorem 2

To evaluate Theorem 2 more comprehensively, below we also plot the $\sum_{c \in \mathcal{C}} p_c$ versus $\frac{|\mathcal{C}|}{k}$ at $k = 0.1|\mathcal{D}|$ and $k = 0.5|\mathcal{D}|$ in Figure 9 and 10, respectively. We also plot $\sum_{c \in \mathcal{C}} p_c$ versus iterations in Figure 11. We summarize that the the column-row index distribution is concentrated on a few column-row pairs. Thus, the assumption in Theorem 2 holds under the context of fine-tuning transformers.

## E.2   More Experimental Speed Analysis

Increasing the batch size can often result in faster model convergence and/or enhance the final performance. Ideally, we should adjust the batch size according to the requirements of our model

|     | Method  | T5-ATT | T5-FF | T5 Block | T5-Large |
|-----|---------|--------|-------|----------|----------|
| Fwd | Full    | 8      | 10    | 17       | 1052     |
|     | WTA-CRS | 22     | 16    | 37       | 2013     |
| Bwd | Full    | 16     | 19    | 34       | 2073     |
|     | WTA-CRS | 15     | 14    | 30       | 1738     |
| F-B | Full    | 24     | 29    | 51       | 3125     |
|     | WTA-CRS | 37     | 30    | 67       | 3751     |

Table 3: Latency (ms) of Forward and Backward pass.

rather than being constrained by the GPU's memory capacity. To illustrate this, we have represented the correlation between peak memory usage and maximum mini-batch size for T5-Base, T5-Large, and T5-3B in Figure 13. Our observations highlight that WTA-CRS effectively increases the maximum available batch size.

We also provide the apple-to-apple speed comparison for linear operation with and without WTA-CRS in Table 3. In Table 3, "Fwd", "Bwd", and "F-B" are the time of forward pass, the time of backward pass, and the total time for both the forward and backward pass, respectively. We summarize that under the same workload, the current implementation of WTA-CRS may roughly slow down the linear operation about 20%. This is because the extra sampling process and data movement counteract the acceleration (see Algorithm 1). However, we note that (1) the overhead can be greatly reduced with better implementation, e.g., using prefetch and operation-fusion technique [30]; (2) the existing implementation can still yield a large speedup when employing larger batch sizes (Figure 8).

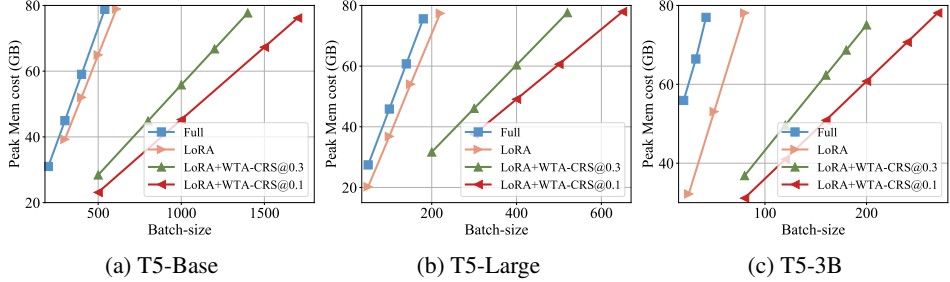

(a) T5-Base        (b) T5-Large        (c) T5-3B

Fig. 13. Peak memory usage versus maximum mini-batch size of T5.

### E.3   Performance of WTA-CRS on OPT models

We conduct experiments of WTA-CRS on OPT models. It is observed in Table 4 that WTA-CRS exhibits almost no drop in accuracy compared to Full training and LoRA on all three architectures. This observation indicates the effectiveness of WTA-CRS, especially considering its consistent performance across the encoder-only, decoder-only and encoder-decoder transformer architectures.

| Method           | CoLA  | MRPC  | RTE   | STS-B | Average |
|------------------|-------|-------|-------|-------|---------|
| Full             | 49.84 | 85.47 | 72.56 | 84.43 | 73.075  |
| LoRA             | 52.3  | 88.36 | 74.01 | 87.21 | 75.47   |
| LoRA+WTA-CRS@0.3 | 51.8  | 88.43 | 74.01 | 86.61 | 75.2125 |

Table 4: Deployment of WTA-CRS@0.3 to OPT-350M.

### E.4   Performance of Activation Quantization+WTA-CRS

To demonstrate the orthogonality of WTA-CRS with activation quantization, we conducted additional experiments of combining WTA-CRS@0.3, along with activation quantization@8bit, on the T5-

base model. The experiment results are presented in Table 5. It is notabe that the combination of LoRA+WTA-CRS@0.3+Quant@8bit exhibits almost no drop in accuracy compared with Full training, LoRA, WTA-CRS@0.3, and LoRA+WTA-CRS@0.3. This observation clearly demonstrates the orthogonality of WTA-CRS with activation quantization, indicating that they can be effectively applied together without compromising performance.

| Method | CoLA | MRPC | RTE | STS-B | Average |
|---|---|---|---|---|---|
| FP32 | 60.1 | 91.5 | 79.4 | 90.6 | 80.4 |
| LoRA | 60.6 | 92.2 | 80.6 | 90.7 | 81.0 |
| WTA-CRS@0.3 | 60.9 | 91.1 | 78.7 | 90.5 | 80.3 |
| LoRA+WTA-CRS@0.3 | 60 | 92 | 80.1 | 90.4 | 80.6 |
| LoRA+WTA-CRS@0.3+Quant@8bit | 60.3 | 92.06 | 81.2 | 90.4 | 81.0 |

Table 5: Combination of `WTA-CRS` with quanzation on the T5-base model.

## E.5  Comparison of `WTA-CRS` with QLoRA

We conduct experiments to compare `WTA-CRS` with QLoRA on the T5-Base model. Specifically, we first quantize the model into NF4 data format using 'bitandbytes' and keep it fixed. Then we apply LoRA+`WTA-CRS`@0.3 over the quantized model. Below in Table 6, we demonstrate that applying QLoRA and `WTA-CRS` over a base model quantized in NF4 results in almost no performance loss against the QLoRA baseline. However, such joint applications may enjoy the training-time memory efficiency offered by WTA-CRS, as illustrated in Table 7.

| Method | Cola | MRPC | RTE | STS-B | Average |
|---|---|---|---|---|---|
| LoRA | 60.6 | 92.2 | 80.6 | 90.7 | 81.025 |
| LoRA+WTACRS@0.3 | 60.0 | 92.0 | 80.1 | 90.4 | 80.625 |
| QLoRA (NF4) | 61.3 | 92.2 | 81.2 | 90.5 | 81.3 |
| QLoRA (NF4)+WTACRS@0.3 | 62.1 | 92.2 | 82.7 | 90.1 | 81.775 |

Table 6: Comparison of `WTA-CRS` with QLoRA.

| Method | Base | Large |
|---|---|---|
| Lora | 13.84 | 36.83 |
| QLora | 13.64 | 36.12 |
| LoRA+WTA-CRS@0.3 | 6.50 | 17.44 |
| QLoRA+WTA-CRS@0.3 | 6.31 | 16.75 |

Table 7: Peak memory usage (GB) of fine-tuning T5-Base with different methods.

## F  Experimental Settings

We give the detailed hyper-parameter setting in this section. Specifically, for both T5 and BERT models, the parameters are updated with the AdamW optimizer $\beta_1 = 0.9$ $\beta_2 = 0.999$ $\epsilon = 10^{-8}$ and weight decay $= 0$. The the learning rate is adjusted with a linear LR Scheduler, which maintains a constant learning rate for the initial 500 steps, and adjusts it gradually thereafter. The input sequences are padded to the maximum length 128. `WTA-CRS` has a LoRA dimension 32 if it is combined with LoRA. To achieve the optimal solution, the T5-Base, Large, 3B and BERT-Base and Large models have different learning rate, training epoch number, and mini-batch size on different datasets, which are given in Tables 9, 10, 11, respectively.

### F.1  Computational Infrastructure

The computational infrastructure information is given in Table 8.

Table 8: Computing infrastructure for the experiments.

| Device Attribute | Value |
|---|---|
| Computing infrastructure | GPU |
| GPU model | NVIDIA-A100 |
| GPU Memory | 81251MB |
| CUDA Version | 11.4 |
| CPU Memory | 512GB |

Table 9: Learning rate.

| Model | Method | CoLA | SST-2 | MRPC | QQP | MNLI | QNLI | RTE | STS-B |
|---|---|---|---|---|---|---|---|---|---|
| BERT-Base | WTA-CRS@0.3 | | | | 2e-5 | | | | |
| | LoRA+WTA-CRS@0.3 | 2e-4 | 5e-4 | 2e-4 | 3e-4 | 3e-4 | 2e-4 | 2e-4 | 3e-4 |
| T5-Base | WTA-CRS@0.3 | | | 3e-5 | | | 3e-6 | 3e-5 | 3e-5 |
| | WTA-CRS@0.1 | | | | 3e-5 | | | | |
| | LoRA+WTA-CRS@0.3 | 3e-4 | 3e-5 | 3e-4 | 3e-5 | 3e-5 | 3e-5 | 3e-4 | 3e-4 |
| | LoRA+WTA-CRS@0.1 | 3e-4 | 3e-5 | 3e-4 | 3e-5 | 3e-5 | 3e-5 | 3e-4 | 3e-4 |
| BERT-Large | WTA-CRS@0.3 | | | | 2e-5 | | | | |
| | LoRA+WTA-CRS@0.3 | 3e-4 | 2e-4 | 2e-4 | 2e-4 | 2e-4 | 2e-4 | 3e-4 | 3e-4 |
| T5-Large | WTA-CRS@0.3 | | | 3e-5 | | | 3e-6 | 3e-5 | 3e-5 |
| | WTA-CRS@0.1 | | | 3e-5 | | | 3e-6 | 3e-5 | 3e-5 |
| | LoRA+WTA-CRS@0.3 | 3e-4 | 3e-5 | 3e-4 | 3e-5 | 3e-5 | 3e-5 | 3e-4 | 3e-4 |
| | LoRA+WTA-CRS@0.1 | 3e-4 | 3e-5 | 3e-4 | 3e-5 | 3e-5 | 3e-5 | 3e-4 | 3e-4 |
| T5-3B | LoRA+WTA-CRS@0.3 | 3e-4 | 3e-5 | 3e-4 | 3e-4 | 3e-4 | 3e-5 | 3e-4 | 3e-4 |
| | LoRA+WTA-CRS@0.1 | 3e-4 | 3e-5 | 3e-4 | 3e-4 | 3e-4 | 3e-5 | 3e-4 | 3e-4 |

Table 10: Training epoch number.

| Model | Method | CoLA | SST-2 | MRPC | QQP | MNLI | QNLI | RTE | STS-B |
|---|---|---|---|---|---|---|---|---|---|
| BERT-Base | WTA-CRS@0.3 | 20 | 20 | 10 | 10 | 10 | 10 | 20 | 10 |
| | LoRA+WTA-CRS@0.3 | 60 | 20 | 20 | 20 | 20 | 20 | 40 | 40 |
| T5-Base | WTA-CRS@0.3 | 40 | 10 | 20 | 10 | 10 | 10 | 50 | 20 |
| | WTA-CRS@0.1 | 40 | 10 | 20 | 10 | 10 | 10 | 50 | 20 |
| | LoRA+WTA-CRS@0.3 | 40 | 10 | 20 | 20 | 20 | 10 | 50 | 20 |
| | LoRA+WTA-CRS@0.1 | 40 | 10 | 20 | 20 | 20 | 10 | 50 | 20 |
| BERT-Large | WTA-CRS@0.3 | 60 | 20 | 20 | 10 | 10 | 10 | 40 | 10 |
| | LoRA+WTA-CRS@0.3 | 60 | 20 | 20 | 20 | 20 | 20 | 40 | 40 |
| T5-Large | WTA-CRS@0.3 | 20 | 10 | 20 | 10 | 10 | 10 | 40 | 20 |
| | WTA-CRS@0.1 | 20 | 10 | 20 | 10 | 10 | 10 | 40 | 20 |
| | LoRA+WTA-CRS@0.3 | 40 | 10 | 40 | 10 | 10 | 10 | 60 | 20 |
| | LoRA+WTA-CRS@0.1 | 40 | 10 | 20 | 10 | 10 | 10 | 60 | 20 |
| T5-3B | LoRA+WTA-CRS@0.3 | 40 | 10 | 20 | 10 | 10 | 10 | 60 | 20 |
| | LoRA+WTA-CRS@0.1 | 40 | 10 | 20 | 10 | 10 | 10 | 60 | 20 |

Table 11: Training mini-batch size. Here for BERT models, we utilize 128 batch size except STS-B, where a batci size 16 is used. For T5 model, we use a 100 batch size across all datasets.

| Model | Method | CoLA | SST-2 | MRPC | QQP | MNLI | QNLI | RTE | STS-B |
|---|---|---|---|---|---|---|---|---|---|
| BERT-Base/Large | WTA-CRS@0.3 | | | | 128 | | | | 16 |
| | LoRA+WTA-CRS@0.3 | | | | 128 | | | | 16 |
| T5-Base/Large/3B | WTA-CRS@0.3 | | | | 100 | | | | |
| | WTA-CRS@0.1 | | | | 100 | | | | |
| | LoRA+WTA-CRS@0.3 | | | | 100 | | | | |
| | LoRA+WTA-CRS@0.1 | | | | 100 | | | | |

