# OpenReview forum: "Winner-Take-All Column Row Sampling for Memory Efficient Adaptation of Language Model"
_NeurIPS.cc/2023/Conference — NeurIPS 2023 poster_

### Official Review · Reviewer_GDNX · 2023-06-29

**Soundness:** 4 excellent
**Presentation:** 3 good
**Contribution:** 3 good
**Rating:** 7
**Confidence:** 4

**Summary:**

>**Rebuttal:** The provided details satisfy my concerns. I think this paper should be accepted after applying the agreed changes.

>**TL;DR:** **Good paper.** The proposed WTA-CRS algorithm is based on the existing CRS algorithm and is used to reduce activation memory during training. WTA-CRS achieves up to 2.7× peak memory reduction with almost no accuracy drop and enables up to 6.4× larger batch size. However, WTA-CRS comes with computational overhead, which is discussed and explore. Addressing my concerns and questions would improve my score.

The paper proposes the WTA-CRS algorithm to reduce the neural networks training activation memory, where the paper claims that activation memory is primary memory bottleneck during training. The WTA-CRS algorithm is an unbiased estimators for matrix production with reduced variance, which only requires storing the sub-sampled activations for calculating the gradient. WTA-CRS achieves up to 2.7× peak memory reduction with almost no accuracy drop and enables up to 6.4× larger batch size.

The WTA-CRS algorithm works by sampling columns and rows to create an unbiased estimation of the original GEMM for the backpropagation. The WTA-CRS algorithm does not alter the neural architecture, and therefore the inference speed is left in tact. The experimental section shows that WTA-CRS outperforms existing prior work and is compatible with existing PEFT techniques. WTA-CRS adds a computational overhead due to sampling, however, WTA-CRS enables training on much larger batch sizes, which results in a 1.2× higher training throughput.


**Strengths:**

* **S.1.** The proposed WTA-CRS algorithm tackles an important problem in existing PEFT techniques, which makes LLM PEFT training more accessible to researchers with low resources.
* **S.2.** The paper provides a theoretical analysis on WTA-CRS.
* **S.3.** The proposed WTA-CRS algorithm outperform existing algorithms.
* **S.4.** An anonymized code repository is provided as part of the submission for reproduction .


**Weaknesses:**

 * **W.1.** Popular existing memory efficient training techniques such as tensor rematerialization (gradient checkpointing) [2][3] and ZeRO [1] are not compared to, although some are partially discussed in Appendix A.
* **W.2.** The experiments are conducted on single neural network architecture (T5), although the proposed technique does not seem to be confined solely to that setting.
* **W.3.** It is common practice today to train neural networks at a lower precision (quantization), however, it is not clear whether quantization (16bit) was used. Therefore, there is insufficient proof that the combined noise of WTA-CRS and quantization would be compatible.


**Typos.**
* Line #62: "Thus" → "Thus,"
* Line #240: "mAccording" → "According"
* Line #297: "Thus" → "Thus,"

[1] Ren, J., Rajbhandari, S., Aminabadi, R.Y., Ruwase, O., Yang, S., Zhang, M., Li, D. and He, Y., 2021, July. ZeRO-Offload: Democratizing Billion-Scale Model Training. In USENIX Annual Technical Conference (pp. 551-564).

[2] Jain, P., Jain, A., Nrusimha, A., Gholami, A., Abbeel, P., Gonzalez, J., Keutzer, K. and Stoica, I., 2020. Checkmate: Breaking the memory wall with optimal tensor rematerialization. Proceedings of Machine Learning and Systems, 2, pp.497-511.

[3] Beaumont, O., Eyraud-Dubois, L. and Shilova, A., 2021. Efficient combination of rematerialization and offloading for training dnns. Advances in Neural Information Processing Systems, 34, pp.23844-23857.

**Questions:**

* **Q.1.** In line #43 and Figure 2 it is noted that "storing activations (or feature maps) is the main memory bottleneck during training". Does this hold true for all model architectures? What about LLM training where the fine-tuning batch size is usually very small?
* **Q.2.** Why was the WTA-CRS algorithm compared to the Deterministic top-k from [1] but not to the Bernoulli-CRS from [1]? What are the key differences between WTA-CRS and Bernoulli-CRS?
* **Q.3.** The paper proposes WTA-CRS which sacrifices computation speed at the cost of lower peak memory. There are several existing common approaches (such as gradient checkpointing and DeepSpeed) for general memory efficient training which are compatible with PEFT techniques. Why are these comparisons not explored or detailed in the main paper?

[1] Adelman, Menachem, Kfir Levy, Ido Hakimi, and Mark Silberstein. "Faster neural network training with approximate tensor operations." Advances in Neural Information Processing Systems 34 (2021): 27877-27889.

**Limitations:**

The limitations are discussed in Appendix A.

---

> ### Author Rebuttal · Authors · 2023-08-09
>
>
> **[W1, Q3] Popular existing memory efficient training techniques such as tensor rematerialization (gradient checkpointing) and ZeRO are not compared to, although some are partially discussed in Appendix A.**
>
> Thank you for the suggestion. We conduct a more detailed comparision between gradient checkpointing and WTA-CRS using huggingface backend: We set 'gradient_checkpointing = True' in the huggingface backend and report the final memory saving.
>
> Table: Comparison between Gradient-checkpoint and WTA-CRS in terms of the memory footprints (GB)
> | Method | T5-Base	| T5-Large |
> | :---: | :---: | :---: |
> | FP	| 17.66	| 45.85
> | Grad-checkpoint	| 13.91 (1.27x)	| 36.5 (1.25x)
> | LoRA+WTA-CRS@0.3	| 8.44 (2.1x)	| 21.58 (2.1x)
> | LoRA+WTA-CRS@0.1	| 7.30 (2.4x)	| 18.46 (2.5×)
>
> Regarding ZeRO [1], up to our knowledge, it is mainly designed for offloading the optimizer states, which is orthogornal to activation memory saving.
>
> **[W2] The experiments are conducted on single neural network architecture (T5)**
>
>
> We appreciate the reviewer for this thoughtful comment. We respectfully point out that we already conducted experiments on **both encoder-only architecture, such as BERT, and encoder-decoder architecture, like T5 in Table 1**.
>
> To further respond to the reviewer's comment, we also conduct additional experiments using the decoder-only architecture, OPT. For your convenience, we summarize the results of these experiments in the below three tables: one each for BERT-Large, OPT-350M, and T5-Large.
>
> We observed that WTA-CRS exhibits almost no drop in accuracy compared to Full training and LoRA on all three architectures. This observation strongly indicates the effectiveness of WTA-CRS, especially considering its consistent performance across diverse transformer architectures.
>
> Table: Encoder-only architecture: BERT-Large
> | Method | CoLA | MRPC | RTE | STS-B | Average |
> | :---: | :---: | :---: | :---: | :---: | :---: |
> | Full	| 66.8	| 89.5	| 72.6	| 90.2	| 79.775
> | LoRA	| 65.9	| 90.8	| 71.3	| 90.3	| 79.575
> | LoRA+WTA-CRS@0.3	| 66	| 89.7	| 72.4	| 89.7	| 79.45
>
> Table: Decoder-only architecture: OPT-350M
> | Method | CoLA | MRPC | RTE | STS-B | Average |
> | :---: | :---: | :---: | :---: | :---: | :---: |
> | Full	| 49.84	| 85.47	| 72.56	| 84.43	| 73.075
> | LoRA	| 52.3	| 88.36	| 74.01	| 87.21	| 75.47
> | LoRA+WTA-CRS@0.3	| 51.8	| 88.43	| 74.01	| 86.61	| 75.2125
>
> Table: Encoder-Decoder architecture: T5-Large
> | Method | CoLA | MRPC | RTE | STS-B | Average |
> | :---: | :---: | :---: | :---: | :---: | :---: |
> | Full	| 61.3	| 93.4	| 85.3	| 91.8	| 82.95 |
> | LoRA	| 63.3	| 93.5	| 84.2	| 91.7	| 83.175 |
> | LoRA+WTA-CRS@0.3	| 62.9	| 93.6	| 83.9	| 91.3	| 82.925
>
>
>
> **[W3] It is not clear whether quantization (16bit) was used. Therefore, there is insufficient proof that the combined noise of WTA-CRS and quantization would be compatible.**
>
> We sincerely appreciate the thoughtful comment provided by the reviewer. We agree with the reviewer's suggestion that including the experiment results of WTA-CRS with bfloat16 quantization is essential to demonstrate its effectiveness under different settings.
>
> To address this concern, we have conducted additional experiments of WTA-CRS@0.3 on the T5-Base model with bfloat16 quantization applied to both the weight and activation map during training. The results of this experiment are presented in the following table. It is revealed that WTA-CRS@0.3 with bfloat16 quantization shows almost no drop in accuracy when compared with Full training and LoRA. This result explicitly demonstrates the effectiveness of WTA-CRS even combined with the bfloat16 quantization, indicating its robustness against the underflow noise caused by bfloat16 quantization.
>
> Table: Accuracy of WTA-CRS@0.3 on T5-Base with bfloat16 quantization.
> | Method | CoLA | MRPC | RTE | STS-B | Average |
> | :---: | :---: | :---: | :---: | :---: | :---: |
> | FP32	| 60.1	| 91.5	| 79.4	| 90.6	| 80.4
> | LoRA-FP32	| 60.6	| 92.2	| 80.6	| 90.7	| 81.0
> | LoRA+WTA-CRS@0.3-FP32	| 60	| 92	| 80.1	| 90.4	| 80.6
> | LoRA+WTA-CRS@0.3-BF16	| 60.3	| 92.4	| 80.1	| 90.19	| 80.7
>
>
> **[Q1] Are the activation memory still the bottleneck for LLM training?**
>
> This is a great question. The short answer is the bottleneck depends on the fine-tuning setting. In data parallel/single GPU training setting, we need to store the model and optimizer states in GPU memory, and the left space is for holding activations. Thus, when the size of LLM goes beyond a certain threshold, the model weight/optmizer state must become the memory bottleneck. This is also why LLM fine-tuning often comes with a small batch size, resulting in a low GPU utility. However, when the size of LLM becomes too large such that it and its optimizer cannot be hold into one single GPU, we must tune it with pipeline/tensor/model parallelism. In such setting, it requires the division of the model into smaller segments, which are then distributed across multiple devices. In this case, each GPU only hold a small part of models, and the space left for activations become much larger. In this case, if we enlarge the sequential length and/or batch size, the activation is still the bottleneck.
>
> **[Q2] Why was the WTA-CRS algorithm compared to the Deterministic top-k from [1] but not to the Bernoulli-CRS from [1]?**
>
> We thank the reviewer for this comment. We compare our algorithm to Deterministic top-k mainly because Deterministic Top-K works better than Bernoulli-CRS  in the context of neural network (Theorem 2 in [2], Figure 1b and Figure 3 in [2]). Thus we follow [2] to compare against top-k CRS instead of Bernoulli-CRS.
>
> **[Typos] Line #62: "Thus" → "Thus,"; Line #240: "mAccording" → "According"; Line #297: "Thus" → "Thus,".**
>
> We appreciate the reviewer for the thoughful comments. We will fix these typos in our camera-ready version.
>
> [1] ZeRO: Memory Optimizations Toward Training Trillion Parameter Models
>
> [2] Faster Neural Network Training with Approximate Tensor Operations

---

> > ### Comment · Reviewer_GDNX · 2023-08-14
> > **Response to Rebuttal**
> >
> > Thank you for the detailed answers and results.
> >
> > The provided results and details satisfy my concerns. I will update my review accordingly.

---

### Official Review · Reviewer_j1mL · 2023-07-04

**Soundness:** 3 good
**Presentation:** 3 good
**Contribution:** 2 fair
**Rating:** 5
**Confidence:** 4

**Summary:**

In this paper, we propose a new method called WTA-CRS (Winner-Take-All Column Row Sampling) to address the main memory bottleneck issue during training, which arises from storing feature maps. To reduce memory usage during training, we sample the most likely column indices during backpropagation.

Furthermore, they proposed method demonstrates the ability to significantly reduce peak memory usage, by approximately up to 2.7 times, when fine-tuning downstream tasks. It also showcases the potential for higher throughput, enabling more efficient training.

**Strengths:**

1. The work clearly states its motivation and its solution and is easy to follow.
2. The authors show that their method reaches comparable performance with backpropagation using the full activation when combined with LoRA.
3. They also empirically measure throughput gains obtained by increasing batch size, which demonstrates the practical applicability of their method.

**Weaknesses:**

1. The paper needs a comparative analysis of other researchs, such as gradient checkpoint/recalculation and CRS, aimed at reducing activation memory during the training phase, as shown in Fig. 6 and Fig. 9.
2. The paper should include an analysis of the overhead associated with the proposed WTS-CRS method, which involves sampling rows and columns. It is crucial to consider factors such as the computational cost of Equation 3 and any potential effects of lowering on the overall performance. Providing this analysis would enhance the clarity and completeness of the research.
3. There is a need of analysis on the effectiveness of the proposed approach, WTS-CRS, in distributed training environments such as tensor parallelism or pipeline parallelism.
4. It seems necessary to conduct performance evaluations on various LLMs of the GPT family, such as LLaMA and OPT.

**Questions:**

* In Figure 9, it can be observed that the throughput of WTS-CRS is lower than that of full when the batch size is small. Is this due to the overhead caused by lowering?
* When comparing the training throughput, how does CRS differ from full in terms of throughput?
* Could the authors include statistics for GPU utilization in their experiments? It would be helpful to analyze the causes of improved performance more thoroughly.
* Considering that most large models are trained using multiple levels of parallelism, would it be possible to verify results for pipeline parallel, tensor parallel, etc.? Also, it is unclear from the paper whether the data parallelism used was distributed data parallelism or naïve data parallelism.

**Limitations:**

* As previously mentioned, it would be valuable to include additional experimental results for models that are more challenging to quantify, such as GPT-series (OPT, LLaMA). This would enhance the validity and applicability of the proposed method across a broader range of models.
* Considering that most large-scale models are trained using multiple levels of parallelism, it is important to assess how much the proposed methods, such as pipeline parallelism and tensor parallelism, can increase throughput while taking into account overhead (such as GPU-to-GPU or node-to-node communication), memory reduction, and computational cost. Furthermore, it is not clear from the paper whether the data parallel processing used is distributed data parallelism or naive data parallelism.

---

> ### Author Rebuttal · Authors · 2023-08-09
>
> **[W1,Q2] Compare against gradient checkpoint/recalculation and CRS**
>
> For the comparison to CRS, from the accuracy perspective, we already compared them in Figure 8. We observe that CRS cannot maintain the accuracy. Thus, it is less applicable to fine-tuning, let alone memory saving. From the memory-saving perspective, CRS and WTA-CRS share the same implementation. Thus if the number of column-row pairs are the same, their memory saving is exactly the same.
>
> For the comparison to gradient checkpoint, **we discussed it in Appendix A**. Here we conduct a more detailed comparison between gradient checkpointing and WTA-CRS within the huggingface backend: We set 'gradient_checkpointing = True' in the configuration and report the final memory saving.
>
> Table: Comparison between gradient checkpoint and WTA-CRS in terms of memory footprints (GB)
> | Method | T5-Base	| T5-Large |
> | :---: | :---: | :---: |
> | FP	| 17.66	| 45.85
> | Grad-checkpoint	| 13.91 (1.27x)	| 36.5 (1.25x)
> | LoRA+WTA-CRS@0.3	| 8.44 (2.1x)	| 21.58 (2.1x)
> | LoRA+WTA-CRS@0.1	| 7.30 (2.4x)	| 18.46 (2.5×)
>
>
> **[W2,Q1] The analysis of the overhead associated with WTS-CRS**
>
> We kindly draw your attention to **Appendix E.2**, where we already conducted an in-depth analysis. We present the table here for your convenience.
>
> The following table provides a breakdown of the latency in our implementation: Fwd', 'Bwd', and 'F-B' represent the time of forward pass, backward pass, and the total time for both the forward and backward pass, respectively. We summarize that, under the same workload, **the current implementation of WTA-CRS may experience a roughly 20% slowdown in linear operation**. This can be attributed to the extra sampling process.
>
> Although we remove 70% column-row pairs, the backward time is only slightly faster than the baseline. This is mainly because the current implementation separately indexes a subset of gradient tensor before multiplying with the subsampled activations, which incurs many extra I/O [2] (also, please check Figure 13 in [2]). **Fortunately, this overhead can be sigificantly reduced with kernel fusion using Triton [3]. According to Figure 13 in [2], we expected a 2X speedup for backward pass with this Tritnon implementation [3].**
>
> Table: Latency (ms) of Forward, Backward and Forward-backward pass.
> | | Method | T5-Attention | T5-FFN | T5-Block | T5-Large |
> | :---: | :---: | :---: | :---: | :---: | :---: |
> | Fwd | Full | 8 | 10 | 17 | 1052
> | Fwd | WTA-CRS@0.3 | 22 | 16 | 37 | 2013
> | Bwd | Full | 16 | 19 | 34 | 2073
> | Bwd | WTA-CRS@0.3 | 15 | 14 | 30 | 1738
> | F-B | Full | 24 | 29 | 51 | 3125
> | F-B | WTA-CRS@0.3 | 37 | 30 | 67 | 3751
>
>
> **[W3, Q4, Limitation2] Evaluate WTS-CRS under tensor parallelism or pipeline parallelism. Also, it is unclear from the paper whether the data parallelism used was distributed data parallelism or naïve data parallelism**
>
> For the model/tensor parallelism setting, first, up to our knowledge, it is rarely used in fine-tuning scenario [4], which is the main focus of this paper. For fine-tuning, the ideal case is to use one single GPU to tune a model as large as possible [4].
>
> Second, the pipeline/model parallelism requires the division of the model into smaller segments, which are distributed across multiple devices. Thus, it requires the communication of activations between consecutive model parts, potentially causing substantial overhead [5]. In this context,  WTA-CRS significantly reduces the communication volume by compressing the activation, thus reducing this overhead [5]. We leave it as future work.
>
> For the "data parallelism" question, the "data parallelism" in this paper refers to **distributed data parallelism.**
>
> **[W4, Limitation1] Conduct performance evaluations on LLMs of GPT family, such as LLaMA and OPT.**
>
> Here we conducted additional experiments of applying WTA-CRS to the OPT model in the below table: LoRA+WTA-CRS@0.3 shows almost no drop in accuracy when compared with full training and LoRA. WTA-CRS has been applied to various transformer architectures including encoder-only (BERT), decoder-only (OPT), and encoder-decoder (T5). With this comprehensive evaluation, we believe that the experiments sufficiently demonstrate the effectiveness of WTA-CRS across diverse transformer architectures.
>
>
> Table: Experiment results on OPT-350M.
> | Method | CoLA | MRPC | RTE | STS-B | Average |
> | :---: | :---: | :---: | :---: | :---: | :---: |
> | Full	| 49.84	| 85.47	| 72.56	| 84.43	| 73.075
> | LoRA	| 52.3	| 88.36	| 74.01	| 87.21	| 75.47
> | LoRA+WTA-CRS@0.3	| 51.8	| 88.43	| 74.01	| 86.61	| 75.2125
>
> **[Q3] Statistics for GPU utilization**
>
> WTA-CRS has extra I/O (the sampling process) in return for reduced computations (FLOPs). Thus, the GPU utility of WTA-CRS is expected to lower than the standard training. Below we measure the GPU utility using torch.cuda.utilization: we presented the GPU utility during training T5-Large. Our result indicates that WTA-CRS reduces 30% GPU utilization of foward pass due to the extra sampling process. However, we note that **GPU utilization cannot reflect the wall-clock speed** : Although the GPU utility of the backward pass in full training is 100%, the wall clock time may still longer than WTA-CRS@0.3 as it requires 70% more FLOPs (workload), especially with kernel fusion implementation [3].
>
> Table: GPU utilization of training T5-Large model on NVIDIA A5000 GPU.
> | Method | Fwd | Bwd | Average |
> | :---: | :---: | :---: | :---: |
> | Full    | 75.2% | 100% | 87.6% |
> | WTA-CRS@0.3	| 40.6% | 100% | 70.3% |
>
> [1] GACT: Activation compressed training for generic network architectures
>
> [2] Deja vu: Contextual sparsity for efficient llms at inference time
>
> [3] https://github.com/FMInference/DejaVu/blob/master/Dejavu/src/ops/triton/gather_gemv.py
>
> [4] QLoRA: Efficient Finetuning of Quantized LLMs
>
> [5] Fine-tuning Language Models over Slow Networks using Activation Quantization with Guarantees

---

> > ### Comment · Reviewer_j1mL · 2023-08-16
> >
> > Thank you for the detailed answers and results.
> >
> > I have read the authors' rebuttal as well as other reviews. I would like to keep my rating.
> > Some of my concerns have been addressed, but i am unsure how memory-efficient the proposed method is compared to other parameter-efficient adaptation techniques like QLoRA[1] and AlphaTuning[2]. Unlike other parameter-efficient adaptation methods, I believe that the approach suggested by the authors may not yield benefits in terms of memory efficiency during the inference process in actual service execution.
> >
> > [1] QLoRA: Efficient Finetuning of Quantized LLMs
> >
> > [2] AlphaTuning: Quantization-Aware Parameter-Efficient Adaptation of Large-Scale Pre-Trained Language Models

---

> > > ### Author Response · Authors · 2023-08-17
> > > **Additional Clarification Regarding Inference Time Memory Efficiency**
> > >
> > > We thank the reviewer for your recognition and active engagement. We find your additional question regarding *inference-time memory efficiency* interesting, though this question might require some background/terminology clarifications to be properly addressed.
> > >
> > > First, we'd like to point out that **classic parameter-efficient fine-tuning (PEFT) techniques can NOT reduce inference memory usage** (e.g., LoRA [1]). However, **such efficiency can indeed be achieved with PEFT utilized in conjunction with model quantization.** For this conversation, we can roughly categorize PEFT techniques into three groups:
> > >
> > > 1. **Full precision PEFT**: where both the base model and the PEFT add-ons are in high FP precision (so no inference memory saving). e.g., LoRA [1] and Adapter tuning [2].
> > > 2. **PEFT utilized with quantization**: where a standard PEFT technique from #1 is applied to a quantized base model. e.g., QLoRA [3], which uses standard LoRA on an NF4-quantized model (with some extra optimization designs engineered).
> > > 3. **Quantization-aware PEFT techniques**: much like #2, but this group of PEFT techniques is designed/adjusted to interfere with the quantization procedure. e.g., AlphaTuning [4], which tunes on the scaling factor of the quantized base model.
> > >
> > > Under this landscape, inference-time memory efficiency can only be gained with PEFT #2&3, where the fine-tuned model is (at least partially) quantized. We argue **by leveraging its orthogonality with QLoRA-like techniques, WTA-CRS may achieve the same goal of delivering a quantized fine-tuned model, thus reducing memory usage during inference** (given the nature of WTA-CRS is a randomized algorithm applicable to any matrix multiplication operations, which happen to be prevalent in LoRA-like setups).
> > >
> > > Below in Table 1, we demonstrate that **applying QLoRA and WTA-CRS over a base model quantized in NF4 results in no performance loss against the QLoRA baseline**. Moreover, such joint applications may enjoy the exciting training-time memory efficiency offered by WTA-CRS (over a naïve QLoRA), as illustrated in Table 2.
> > >
> > >
> > > Table 1: Apply WTA-CRS over quantized T5-Base in 4-bit NormalFloat (NF4) data format using `bitandbytes` [5].
> > > |                         | Cola  | MRPC  | RTE   | STS-B | Average |
> > > | :---------------------: | :---: | :---: | :---: | :---: | :-----: |
> > > | LoRA                    | 60\.6 | 92\.2 | 80\.6 | **90\.7** | 81\.025 |
> > > | LoRA + WTA-CRS@0\.3        | 60    | 92    | 80\.1 | 90\.4 | 80\.625 |
> > > | QLoRA (NF4)             | 61\.3 | **92\.2** | 81\.2 | 90\.5 | 81\.3   |
> > > | [NEW] QLoRA (NF4) + WTA-CRS@0\.3 | **62\.1** | **92\.2** | **82\.7** | 90\.1 | **81\.775** |
> > >
> > > Table 2: Peak memory usage (GB) of fine-tuning T5-Base and T5-Large with different methods.
> > > | Method | Base	| Large|
> > > | :---: | :---: | :---: |
> > > | LoRA	| 13.84 	| 36.83 |
> > > | QLoRA	| 13.64 	| 36.12 |
> > > | LoRA + WTA-CRS@0.3	|  6.50	| 17.44|
> > > | QLoRA + WTA-CRS@0.3	|  **6.31**	| **16.75** |
> > >
> > > We believe our added experiments/discussion justified the soundness of WTA-CRS in regards to inference memory efficiency, and we hope the reviewer may consider raising the score should you find it the same way, or specify what else we can offer to facilitate your judgment.
> > >
> > > ---
> > >
> > > [1] Hu & Shen et al., LoRA: Low-Rank Adaptation of Large Language Models. ICLR 2022
> > >
> > > [2] Houlsby et al., Parameter-Efficient Transfer Learning for NLP. ICML 2019
> > >
> > > [3] Dettmers & Pagnoni et al., QLoRA: Efficient Fine-tuning of Quantized LLMs. arXiv 2023
> > >
> > > [4] Kwon et al., AlphaTuning: Quantization-Aware Parameter-Efficient Adaptation of Large-Scale Pre-Trained Language Models. EMNLP 2022
> > >
> > > [5] https://github.com/TimDettmers/bitsandbytes

---

> > > > ### Author Response · Authors · 2023-08-21
> > > > **Further Discussion with Reviewer j1mL**
> > > >
> > > > Right now, with `7764`, you are the only reviewer staying on the negative side of the rating. We greatly appreciate your feedback and want to ensure we have comprehensively addressed all your concerns. Here, we provide a contextual summary:
> > > >
> > > > Your initial requests can be described as follows:
> > > > 1. a comparison to gradient checkpoint/recalculation and CRS;
> > > > 2. the speed overhead analysis of CRS;
> > > > 3. the ablation study on decoder-only architecture
> > > > 4. evaluate our method under tensor/pipeline parallelism.
> > > >
> > > > In our first-round [rebuttal](https://openreview.net/forum?id=SquMNyrk1O&noteId=C8D3lWSbd2), we have directly addressed concerns (1)(2)(3) by presenting additional experimental results, **which are all supportive to our proposed approach.** For (4), we indicated that this is a less-common setting in fine-tuning and analyzed why tensor/pipeline parallelism could benefit from activation compression.
> > > >
> > > > After the first round rebuttal, your remaining concern is regarding our method's ability to achieve [inference time memory efficiency](https://openreview.net/forum?id=SquMNyrk1O&noteId=4PlMMl78l7) when compared to other quantized PEFTs like QLoRA and AlphaTuning. In our follow-up [response](https://openreview.net/forum?id=SquMNyrk1O&noteId=ctIaP5BbT5), we believe we have clearly justified how and why our method could also improve inference time memory efficiency. Namely, while our method *by itself* — a randomized matrix multiplication algorithm — cannot directly obtain inference-time benefits, we argue that through the orthogonality with QLoRA-like techniques, **our method can achieve the same goal of delivering a quantized fine-tuned model, thus gaining memory efficiency during inference (while enjoying a larger batch size during fine-tune).** Additional results also support our claims.
> > > >
> > > > We understand the reviewer may have multiple papers to review and might need to attend to personal matters, especially over the weekend. However, we believe the remaining concern has been well-discussed and, therefore, hopefully addressed. If the reviewer feels our [last reponse]((https://openreview.net/forum?id=SquMNyrk1O&noteId=ctIaP5BbT5)) clarifies and addresses your remaining concerns, we kindly ask you to consider raising the score to provide a more up-to-date reflection of our work.
> > > >
> > > > Best,
> > > >
> > > > Authors

---

### Official Review · Reviewer_cMiu · 2023-07-06

**Soundness:** 2 fair
**Presentation:** 3 good
**Contribution:** 3 good
**Rating:** 6
**Confidence:** 4

**Summary:**

The authors studied fine-tuning LLMs with limited memory. As the increased scale of current LLMs, the memory cost during fine-tuning is of great importance when adapting the pretrained LLMs to down-streaming tasks. In contrast to the existing work that mainly focus on the number of updated weights, this paper proposed to reduce the number of stored activations, also the inputs to each layer. Given the widely used stochastic gradient descent optimization pipeline, the authors proposed to store a subset of activations that can generate an unbiased gradient estimation. This way, the training memory and the training time decreased significantly. The authors provide both theoretical and experimental analysis on their CRS methods.

**Strengths:**

- This paper studied an important problem in LLM fine-tuning, i.e., how to fine-tuning LLMs with less memory consumption without increasing the computation cost. The authors provided solid quantitative results to show that the main memory consumption is from storing the intermediate activations.
- The authors provided a general solution for fine-tuning LLMs under memory constraints. The solution can be applied in most transformer-based network architectures.
- The authors provided solid mathematical proof on the unbiased gradient estimation, which is especially encouraged.
- The extensive experiments on different network architectures showed the efficacy of the methods.
- The released code can benefit the following researchers studying efficient LLM fine-tuning.

**Weaknesses:**

- I am not fully convinced by the comment made in Line241-244, i.e., the methods in the paper is orthogonal to the activation quantization. When activation is quantized into a lower bit width, it is very possible that the number of less important activations will decrease. This way, the selection on the top-k columns in activation matrices with the proposed methods may hurt the training accuracy or convergence. It would be great if the authors can provide some theoretical analysis or experimental results on this combination. Otherwise, it would be necessary to provide some comparison results w.r.t. the activation quantization.
- It would be great if the authors can discuss the main difference of their paper w.r.t. [Randomized Automatic Differentiation, ICLR2021].

**Questions:**

Overall, I think this paper has a relatively high quality in both writing and scientific contribution.

**Limitations:**

Yes

---

> ### Author Rebuttal · Authors · 2023-08-09
>
> **[W1] Whether WTA-CRS is compatiable with activation quantization or not** (experiment done)
>
>
> We thank the reviewer for this thoughtful comment. We acknowledge the importance of demonstrating the orthogonality of WTA-CRS with activation quantization. To address this concern, we conducted additional experiments of combining WTA-CRS@0.3, along with activation quantization@8bit, on the T5-base model.
>
> The experiment results are presented in the following table. It is notabe that the combination of LoRA+WTA-CRS@0.3+Quant@8bit exhibits almost no drop in accuracy compared with Full training, LoRA, WTA-CRS@0.3, and LoRA+WTA-CRS@0.3. This observation clearly demonstrates the orthogonality of WTA-CRS with activation quantization, indicating that they can be effectively applied together without compromising performance.
>
> Table: Combination of WTA-CRS with quanzation on the T5-base model.
> | Method | CoLA | MRPC | RTE | STS-B | Average |
> | :---: | :---: | :---: | :---: | :---: | :---: |
> | Full | 60.1	| 91.5	| 79.4	| 90.6	| 80.4 |
> | LoRA |60.6	|92.2	|80.6	|90.7	|81.0 |
> | WTA-CRS@0.3 |60.9	|91.1	|78.7	|90.5	|80.3 |
> | LoRA+WTA-CRS@0.3 |60     | 92	|80.1	|90.4	|80.6 |
> | LoRA+WTA-CRS@0.3+Quant@8bit |60.3	|92.06	|81.2	|90.4	|81.0|
>
>
>
>
> **[W2] Discuss the difference about WTA-CRS and RAD**
>
> WTA-CRS and RAD shares the same spirit in the sense that they both trade gradient noise in return for reduced memory. However, the main difference between them lies in how they generate the noisy gradient. Specifically, WTA-CRS focues on approximating expensive matrix production operation. RAD proposes two noisy-yet-cheap gradient estimator, i.e. path sampling (sampling the computation path) and random matrix injection (apply random projection to activations). These techniques are orthogonal to each other. We will include this discussion in the updated version.

---

> > ### Comment · Reviewer_cMiu · 2023-08-14
> >
> > I have read the authors' rebuttal as well as other reviews. I would like to keep my rating.

---

### Official Review · Reviewer_ky3t · 2023-07-06

**Soundness:** 4 excellent
**Presentation:** 3 good
**Contribution:** 3 good
**Rating:** 7
**Confidence:** 3

**Summary:**

The paper's contribution is in proposing a practical, intuitive yet not trivial unbiased approximation to gradient training of matrix multiplication. It shows that even though totally deterministic sampling is biased, somewhat deterministic sampling is unbiased, and a judicious allocation of sampling to those pairs favored by deterministic thinking can lead to the use of a larger batch size with empirically negligible performance loss. This reviewer must declare that he does not check the derivation very carefully.

**Strengths:**

The proposed idea is practical and can be readily combined with virtually all first-order gradient-based training methods.
The paper also derived why deterministic sampling is a biased estimator and empirically shown the associated bad performance, thus proving that the additional complexity of stochastic sampling over deterministic sampling is not only sufficiently better but also necessary.

**Weaknesses:**

It's just a few empirical comparisons, but the performance gap between CRS and WTA-CRS seems modest.

**Questions:**

This reviewer does not have a question.

---

> ### Author Rebuttal · Authors · 2023-08-09
>
> Thank you for spending time and effort in reviewing our paper. We appreciate your constructive comments and suggestions for improving the quality of this work. The feedback truly encouraged us to increase our efforts in conducting quality and impactful research.

---

> > ### Author Response · Authors · 2023-08-16
> > **Authors' clarification**
> >
> > **The performance gap between CRS and WTA-CRS seems modest.**
> >
> > We sincerely apologize that we previouly forgot to reply this weakness. Here we would like to clarify that (1)  we theoretically and empirically show WTA-CRS has smaller variance than CRS. (2) the performance gap seems modest due to the extended y-axis range of Figure 8. We summarize their performance gap from Figure 8 to the below table. We can observe that their performance gap is about 1-3%, which shows the effectiveness of WTA-CRS.
> >
> > | Method | SST2 | MNLI | QQP | Average |
> > | :---: | :---: | :---: | :---: | :---: |
> > | CRS@0.1 | 93.9 $\pm$ 0.1	|82.2 $\pm$ 0.05	|85.5 $\pm$ 0.2	|87.2 $\pm$ 0.1	|
> > | WTA-CRS@0.1 |94.7 $\pm$ 0.01     | 85.3 $\pm$ 0.01	|86.7 $\pm$ 0.1	|88.9 $\pm$ 0.04	|

---

### Author Rebuttal · Authors · 2023-08-09

We thank all the reviewers for their constructive comments and helpful feedback. We are encouraged to find that they have found our contributions to be technically solid (ky3t, cMiu, GDNX), timely and relevant for LLM research (cMiu, GDNX), mathematically solid (cMiu, GDNX), and easy-to-follow (cMiu, j1mL, GDNX).

We have additionally performed experiments to address some of the evaluation concerns. Please find below our detailed response to the questions and any concern raised by the reviewers. We will incorporate all these comments and comprehensive experimental evaluations into the revised manuscript. We are grateful to the reviewers for all the suggestions to improve our work.

Best regards,

Authors

## Summary of Rebuttal

We thank all the reviewers for their constructive comments and helpful feedback. We value their comments sincerely, and do our best to address the concerns. During the rebuttal, we provide the following new supplementary results and analysis:

- (cMiu) We have conducted an additional experiment that combines WTA-CRS with activation map quantization. [[Redirection]](https://openreview.net/forum?id=SquMNyrk1O&noteId=S24PVc9nC9)
- (cMiu, GDNX) We provide a detailed discussion on the distinctions between WTA-CRS and RAD [[Redirection]](https://openreview.net/forum?id=SquMNyrk1O&noteId=S24PVc9nC9), as well as ZeRO [[Redirection]](https://openreview.net/forum?id=SquMNyrk1O&noteId=3w86hHloPJ).
- (j1mL) We present a technical comparison of WTA-CRS with CRS and Gradient-checkpointing, focusing on accuracy and memory cost.  [[Redirection]](https://openreview.net/forum?id=SquMNyrk1O&noteId=C8D3lWSbd2)
- (j1mL) We emphasize the importance of Appendix E.2 in our paper, as it addresses the overhead analysis of WTA-CRS, as requested by the reviewer. [[Redirection]](https://openreview.net/forum?id=SquMNyrk1O&noteId=C8D3lWSbd2)
- (j1mL) We conduct an in-depth analysis of the performance of WTA-CRS in distributed training environments. [[Redirection]](https://openreview.net/forum?id=SquMNyrk1O&noteId=C8D3lWSbd2)
- (j1mL) We provide statistics for GPU utilization to offer further insights into the efficiency of WTA-CRS. [[Redirection]](https://openreview.net/forum?id=SquMNyrk1O&noteId=C8D3lWSbd2)
- (j1mL, GDNX) We include an additional architecture, OPT-350M, which is a decoder-only transformer, in our experiments, to demonstrate the effectiveness of WTA-CRS. [[Redirection]](https://openreview.net/forum?id=SquMNyrk1O&noteId=3w86hHloPJ)
- (GDNX) We have conducted an additional experiment involving the deployment of WTA-CRS in bfloat16 fine-tuning. [[Redirection]](https://openreview.net/forum?id=SquMNyrk1O&noteId=3w86hHloPJ)

---

### Decision · Program_Chairs · 2023-09-21

**Decision:**

Accept (poster)

**Comment:**

The paper presents an interesting novel use use of approximate matrix multiplication to parameter-efficient tuning of LLMs. The approach is theoretically-motivated, and brings significant practical memory improvements.
Given that the reviewer feedback is (almost) uniformly positive, and the authors have committed to several small improvements over the discussion period, providing significant additional data, I am happy to recommend acceptance.